# Persistent directional growth capability in *Arabidopsis thaliana* pollen tubes after nuclear elimination from the apex

Kazuki Motomura [1,2,3], Hidenori Takeuchi [2,4], Michitaka Notaguchi[2,5], Haruna Tsuchi[6], Atsushi Takeda [1,7], Tetsu Kinoshita [6], Tetsuya Higashiyama [2,8,9] & Daisuke Maruyama [6✉]

During the double fertilization process, pollen tubes deliver two sperm cells to an ovule containing the female gametes. In the pollen tube, the vegetative nucleus and sperm cells move together to the apical region where the vegetative nucleus is thought to play a crucial role in controlling the direction and growth of the pollen tube. Here, we report the generation of pollen tubes in *Arabidopsis thaliana* whose vegetative nucleus and sperm cells are isolated and sealed by callose plugs in the basal region due to apical transport defects induced by mutations in the WPP domain-interacting tail-anchored proteins (WITs) and sperm cell-specific expression of a dominant mutant of the CALLOSE SYNTHASE 3 protein. Through pollen-tube guidance assays, we show that the physiologically anuclear mutant pollen tubes maintain the ability to grow and enter ovules. Our findings provide insight into the sperm cell delivery mechanism and illustrate the independence of the tip-localized vegetative nucleus from directional growth control of the pollen tube.

[1] Ritsumeikan Global Innovation Research Organization, Ritsumeikan University, Kusatsu, Shiga 525-8577, Japan. [2] Institute of Transformative Bio-Molecules (WPI-ITbM), Nagoya University, Furo-cho, Chikusa-ku, Nagoya, Aichi 464-8601, Japan. [3] JST, PRESTO, Kawaguchi, Saitama 332-0012, Japan. [4] Institute for Advanced Research, Nagoya University, Furo-cho, Chikusa-ku, Nagoya, Aichi 464-8601, Japan. [5] Bioscience and Biotechnology Center, Nagoya University, Furo-cho, Chikusa-ku, Nagoya, Aichi 464-8601, Japan. [6] Kihara Institute for Biological Research, Yokohama City University, Maioka-cho, Totsuka-ku, Yokohama, Kanagawa 244-0813, Japan. [7] College of Life Sciences, Ritsumeikan University, Kusatsu, Shiga 525-8577, Japan. [8] Division of Biological Science, Graduate School of Science, Nagoya University, Furo-cho, Chikusa-ku, Nagoya, Aichi 464-8602, Japan. [9] Department of Biological Sciences, Graduate School of Science, The University of Tokyo, Hongo, Bunkyo-ku, Tokyo 113-0033, Japan. ✉email: dmaru@yokohama-cu.ac.jp

In angiosperms, the pollen grain is the male gametophyte and consists of two sperm cells within a larger vegetative cell, which produces a pollen tube[1]. The tip-growing pollen tube transports the cytoplasm and sperm cells to the apical region through the periodical production of thick callose plugs, unique cell wall structures enriched in callose (β-1,3 glucan)[2]. Deep inside the pistil, the pollen tube perceives attractant peptides at the tip region and precisely targets the ovule[3], where two sperm cells are released from the pollen tube in one of the two synergid cells and fertilize the egg cell and the central cell[4]. The apical transport of the two sperm cells is a prerequisite for these double fertilization events.

Although the pollen tubes produce callose-enriched cell walls, the pair of sperm cells are generally devoid of conventional cell walls. Sperm cells are wholly enclosed by an endocytic membrane called the inner vegetative plasma membrane and a long tail-like protrusion from a sperm cell is tethered to the vegetative nucleus to form an assemblage termed the male germ unit[5]. During pollen-tube growth in *Arabidopsis thaliana*, the vegetative nucleus enters the pollen tube before the two sperm cells. Klarsicht/ANC-1/Syne-1 Homology (KASH) proteins such as WPP domain-interacting tail-anchored protein 1 (WIT1) and WIT2 are located on the outer nuclear envelope and are involved in the transport of the vegetative nucleus[6]. In the *wit1 wit2* double mutant, the relative positions of the vegetative nucleus and sperm cells are inverted compared with those in the wild-type, and the vegetative nucleus appears to be dragged by the sperm cells. The phenotype of this double mutant suggests the existence of the putative KASH-independent transport of sperm cells[6]. However, the nature of this mechanism remains poorly understood.

Transcriptome and translatome data have demonstrated considerable changes in gene expression patterns after the production of pollen tubes[7–10]. The genes specifically expressed in the pollen tubes include those that code for receptors, ligands, and enzymes that are indispensable for active tip growth as well as communication with female gametophytic cells. In *A. thaliana*, mutants lacking sperm cells do not affect the growth control and discharge of the pollen tube, supporting the idea that the vegetative nucleus and its transcriptional activity regulate pollen-tube growth and functions[11,12].

Here, we report that the over-accumulation of callose through the expression of the dominant mutant of CALLOSE SYNTHASE 3 from a sperm cell-specific promoter could induce severe defects in the apical transport of sperm cells. By introducing this fusion gene into the vegetative nucleus-immotile *wit1 wit2* double mutant, we generated pollen tubes whose vegetative nuclei and sperm cells were retained in the grain and separated from the growing apical region after the formation of a callose plug. Despite this physiologically anuclear condition, the pollen tube exhibited normal directional growth behavior. Our data not only supported sperm cell-specific motive force, but also demonstrated the independence of vegetative nucleus position from various pollen-tube functions.

## Results

### Callose accumulation on the sperm cell periphery reduced male fertility.
Callose is involved in various cellular events, such as cell plate formation, plasmodesmata-mediated protein/RNA transport, response to wounding stress, and pollen-tube growth[13–16]. To test the importance of the absence of cell wall components around sperm cells, we induced ectopic callose deposition by expressing a putative constitutively active mutant of CALLOSE SYNTHASE 3 (*cals3m*)[17]. In the wild-type *A. thaliana* pollen, no callose accumulation was observed after aniline blue staining (Fig. 1a). However, approximately half of the pollen from

transgenic lines hemizygous for *cals3m* under the control of the sperm cell-specific *HISTONE THREE RELATED 10* (*pHTR10: cals3m*, designated hereafter *SC-cal*) displayed callose deposition around the sperm cells (Fig. 1b)[18]. To examine fertility, we performed a reciprocal cross experiment. When pistils from five independent *SC-cal* $T_1$ lines were fertilized by wild-type pollen, four of these lines showed ~50% hygromycin resistance in their $F_1$ siblings (Fig. 1c), which is indicative of the normal inheritance of the *SC-cal* transgenes from female gametophytes. In contrast, hygromycin resistance in the $F_1$ siblings was less than ~25% after cross-pollination between wild-type pistils and pollen from the *SC-cal* transgenic lines (Fig. 1d). We observed a full set of seeds in the siliques of wild-type plants (Fig. 1e), whereas the *SC-cal* transgenic lines showed aborted ovules in their siliques (Fig. 1f). We concluded that ectopic callose deposition on sperm cells caused partial male sterility.

To investigate the callose-induced fertilization defect, we introduced *SC-cal* with a constitutive nuclear marker *pRPS5A: HISTONE 2B-tdTomato* (*RHT*)[19] and selected new hemizygous *A. thaliana* plants displaying genetically linked callose accumulation and nuclear tdTomato signal (Fig. 2a; hereafter designated as *SC-cal RHT* hemizygous plants). In the *SC-cal RHT* hemizygous plants, we did not observe any defects during pollen formation (Supplementary Fig. 1). To examine whether callose deposition causes abnormalities in gene expression and protein localization to the plasma membrane in sperm cells, we generated *SC-cal RHT* hemizygous plants that were homozygous for the *pGCS1:GCS1-Clover*, which is a translational Clover fusion reporter of sperm cell-specific plasma membrane protein *GENERATIVE CELL-SPECIFIC 1* (*GCS1*)/*HAPLESS2*[20,21]. The Clover signal was observed in ~80% of pollen irrespective of whether the pollen was tdTomato-negative (*SC-cal − RHT −*) or tdTomato-positive (*SC-cal + RH +*) (Fig. 2b–d), indicating that callose deposition around sperm cells had little or no effect on their differentiation.

The alteration of the sperm cell wall by callose accumulation might affect the inner vegetative plasma membrane enclosing two sperm cells (Fig. 2a)[1,5,6]. Thus, we generated *SC-cal RHT* hemizygous plants that were homozygous for *pACA3:Lyn24-mNeonGreen*, which expresses mNeonGreen fused with the 24 N-terminus amino acids of *Mus musculus* Lyn protein (Lyn24) from a vegetative cell-specific *ACA3* promoter[22]. The Lyn24 contains the targeting signals of both myristoylation and palmitoylation; the GFP-fused Lyn24 has been previously shown to label a structure with an eyeglass-shaped outline corresponding to the inner vegetative plasma membrane[23]. Such eyeglass-shaped pattern was observed in 76.9% of the *SC-cal − RHT −* pollen (Fig. 2e, g). In contrast, the mNeonGreen signal was uniformly diffused and the frequency of the eyeglass-shaped pattern decreased to 30.8% in tdTomato-positive *SC-cal* pollen (Fig. 2f, g). These data implied the possible alteration of the outer environment of the sperm cell by callose accumulation.

### Apical transport defects were observed in callose-accumulating sperm cells.
To analyze the origin of the partial male sterility of the pollen with callose around sperm cells, we observed in vitro-germinated pollen tubes from the *SC-cal RHT* hemizygous plant shown in Fig. 2a. In wild-type pollen tubes from the *RHT* transgenic line, two sperm nuclei and one vegetative nucleus were observed in the apical region (Fig. 3a). However, in the pollen tubes from the *SC-cal RHT* hemizygous plants, sperm cells were surprisingly present in the basal part of the pollen tube, behind the vegetative nucleus relative to the apical region (Fig. 3b). To further investigate this pollen-tube phenotype, wild-type pistils pollinated with the *SC-cal RHT* hemizygous plant were dissected and incubated on a pollen-tube growth medium containing the

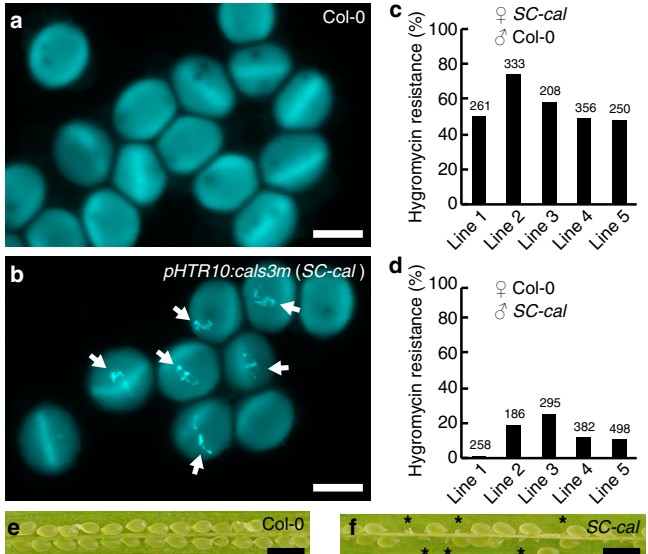

**Fig. 1 Partial male sterility induced by callose accumulation in sperm cells. a**, **b** Aniline blue staining of mature pollen from wild-type Col-0 plants (Col-0, **a**) or hemizygous *pHTR10:cals3m* plants (*SC-cal*, **b**). Arrows in (**b**) indicate callose deposition in sperm cells. Images are representative of three (**a**) or six (**b**) observations. **c**, **d** Transmission of *SC-cal* transgene from female gametophytes (**c**) or male gametophytes (**d**) analyzed by hygromycin resistance of T₁ plants obtained following reciprocal crosses between the *SC-cal* transgenic lines and wild-type Col-0 plants. Line 2 shows higher hygromycin resistance in (**c**) possibly due to the multiple insertions of the *SC-cal*. Numbers above the bars indicate seeds analyzed. **e**, **f** Developing seeds in Col-0 (**e**) or an *SC-cal* transgenic line (**f**). Images in (**e**) or (**f**) show representative of two pistils analyzed. Asterisks in (**f**) represent undeveloped seeds. Scale bars: 20 μm in (**b**, **c**); 1 mm in (**e**, **f**).

nuclear staining dye SYBR Green I. Then, pollen tubes that emerged from the cut styles were analyzed using confocal laser scanning microscopy (Fig. 3c, e). In tdTomato-negative pollen tubes (*SC-cal − RHT −*), nuclear triplets of the vegetative nucleus and sperm nuclei were observed at the apical region of the pollen tubes (Fig. 3d, e, Normal). However, among the 81 pollen tubes labeled with both tdTomato and SYBR Green I in the vegetative nucleus (*SC-cal +*, *RHT +*), 69% did not contain sperm nuclei in the apical region (Fig. 3d, e, VN only). Although most of the remaining *SC-cal + RHT +* pollen tubes (25%) had three nuclei, sperm cells were further away from the vegetative nucleus than *SC-cal − RHT −* pollen tubes (Fig. 3d, e, Long VN-SN and Supplementary Fig. 2). Time-lapse imaging and kymographs in *SC-cal − RHT −* pollen tubes showed that sperm nuclei maintained 5–10 μm intervals from the vegetative nucleus. However, in the *SC-cal + RHT +* pollen tubes, the distance between the vegetative nucleus and the sperm nuclei fluctuated (Fig. 3f–i and Supplementary Movie 1). The unstable nuclear positions in *SC-cal + RHT +* could result from the loss of physical tethering between the vegetative nucleus and sperm cells. This hypothesis was confirmed using the nuclear envelope reporter *pRANGAP1: RANGAP1-mNeonGreen* to visualize the physical tethering (Supplementary Movie 2)[6].

To analyze the effect of callose production around sperm cells on pollen-tube discharge, wild-type pistils were pollinated with pollen from the *SC-cal RHT* hemizygous plant, and Congo red staining was performed to elucidate pollen-tube growth patterns 16 h after pollination (Fig. 4). We found that 92.6% of ovules received one or two pollen tubes (*n* = 498). Therefore, pollen-tube attraction by the ovules appeared to be normal.

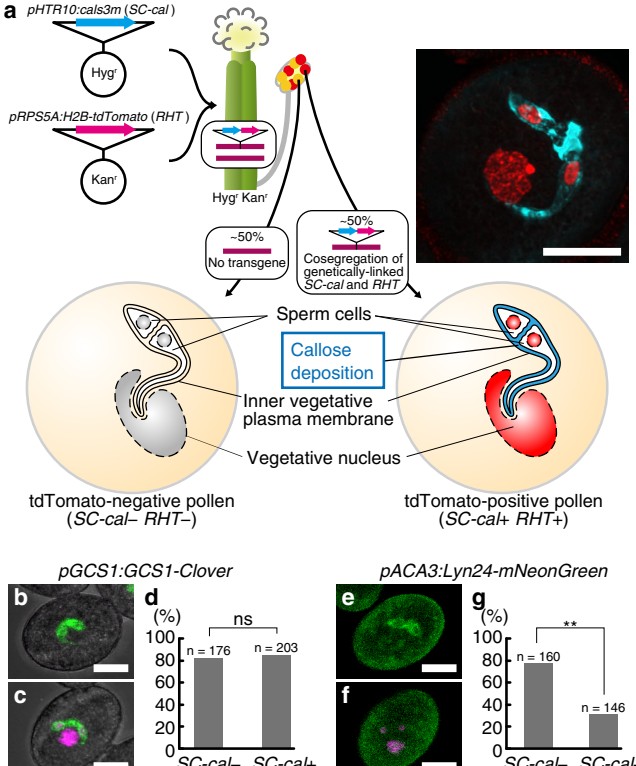

**Fig. 2 Callose accumulation causes severe defect in sperm cell migration. a** Generation of hemizygous plants exhibiting genetically linked *pHTR10:cals3m* (*SC-cal*) and *pRPS5A:H2B-tdTomato* (*RHT*) nuclear marker. The right panel is representative of two independently obtained super-resolution images of aniline-blue-treated pollen from the *SC-cal RHT* line. **b–d** Expression of sperm cell-specific *pGCS1:GCS1-Clover* reporter gene in the *SC-cal RHT* hemizygous plant. The percentages of Clover-labeled sperm cells in tdTomato-negative pollen (*SC-cal − RHT −*, **b**) and tdTomato-positive pollen (*SC-cal + RHT +*, **c**) are shown in (**d**). *P* = 0.76. **e–g** Localization of an inner vegetative plasma membrane marker protein, Lyn24-mNeonGreen, in the pollen from a *SC-cal RHT* hemizygous plant. Percentages of eyeglass-shaped mNeonGreen signal in tdTomato-negative pollen (*SC-cal − RHT −*, **e**) and tdTomato-positive pollen (*SC-cal + RHT +*, **f**) are shown in (**g**). *P* = 4.04 × 10⁻⁸. Scale bar: 10 μm. Asterisks, significance by Pearson's chi-square test. ns, not significant.

When the *SC-cal − RHT −* wild-type pollen tubes reach the ovules and release sperm cells and vegetative nucleus, the ovules show no male-derived tdTomato signal around the micropyle and terminate further pollen-tube attraction following double fertilization[24]. Consistently, 45.6% of the ovules received a single pollen tube and no tdTomato signals were observed (Fig. 4a). Conversely, ovules receiving single pollen tubes often had a single focus of the vegetative nucleus (25.9%, Fig. 4b), but rarely contained multiple tdTomato-labeled nuclei (2.2%, Fig. 4c). After the reception of the *SC-cal + RHT +* pollen tube, double fertilization could not occur because two sperm cells were not present. We also observed ovules with the characteristic physiological response of fertilization failure marked by multiple pollen-tube insertions resulting from the persistent attraction of the pollen tube after the reception of a fertilization-defective pollen tube (15%, Fig. 4e, f)[24], partial seed coat formation, and seed abortion (Supplementary Fig. 3)[25]. We concluded that the loss of sperm cell transport, induced by callose accumulation, was the primary reason for the partial male sterility observed in the *SC-cal* line.

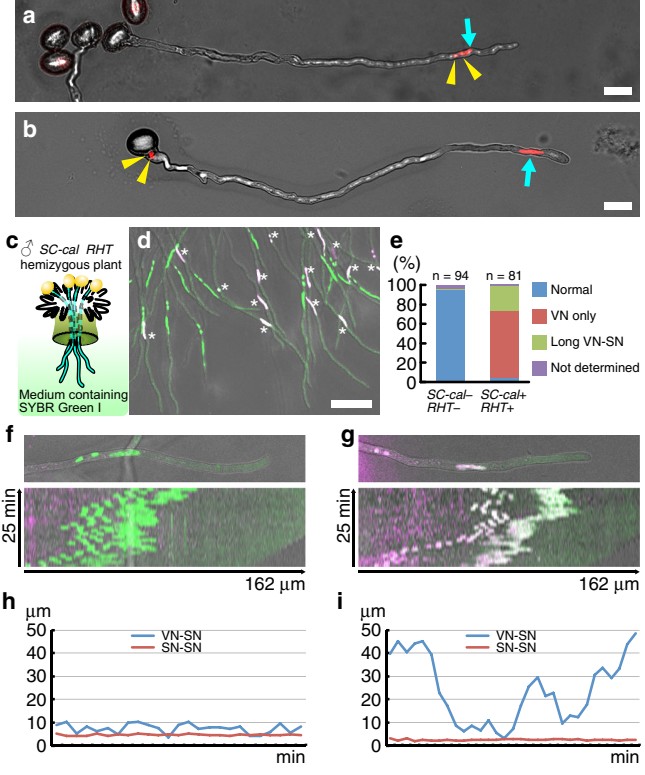

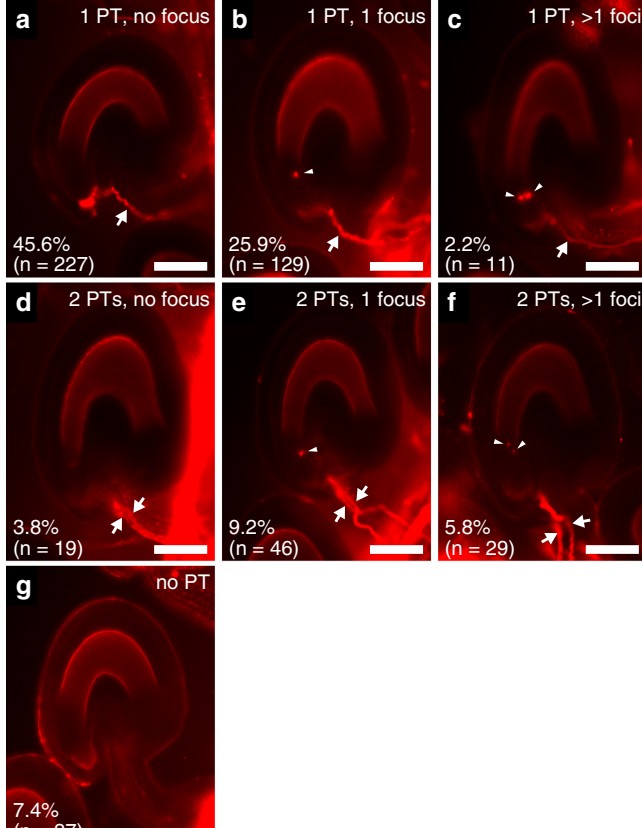

**Fig. 3 Aberrant apical transport of the callose-accumulating sperm cells.**
**a** In vitro-germinated pollen tube from a plant transformed with only *pRPS5A: H2B-tdTomato* (*RHT*) nuclear marker. **b** In vitro-germinated pollen tube from a double transgenic plant harboring *pHTR10:cals3m* (*SC-cal*) and *RHT* (*SC-cal RHT* hemizygous plant identical to the line used in Fig. 2). Yellow arrowheads and cyan arrows represent the sperm nuclei and vegetative nuclei, respectively. Representative images obtained using eleven (**a**) or seven (**b**) independent assays are shown. Scale bar: 20 μm. **c** Schematic drawings of the semi-in vitro pollen-tube growth assay. **d–i** Semi-in vitro pollen-tube growth assay of the *SC-cal RHT* hemizygous plant on a medium containing SYBR Green I for nuclear staining. Apical region of pollen tubes containing tdTomato-negative nuclei (*SC-cal − RHT −*, green nuclei) or tdTomato-positive nuclei (*SC-cal + RHT +*, magenta or white nuclei indicated by asterisk) in a representative image obtained from three independent assays (**d**). Frequency of nuclear patterns (**e**). Note that vegetative nuclei with no sperm cell around (VN only) or apart from two sperm cells (Long VN-SN) were frequently observed in the *SC-cal + RHT +* pollen tubes. Scale bar: 50 μm. Representative images of the *SC-cal − RHT −* (**f**) or *SC-cal + RHT +* (**g**) pollen tubes with kymographs of nuclear movement. The vertical and horizontal axes represent time and distance along the pollen tube, respectively. Transitions of distances from the vegetative nucleus to the front sperm nucleus (VN–SN) or the distance between two sperm nuclei (SN–SN) in (**f** and **g**) were analyzed in (**h** and **i**), respectively.

**Fig. 4 Analysis of pollen-tube behavior in wild-type pistils. a–g** Congo red pollen-tube staining of wild-type ovules 16 h after pollination of pollen from a double transgenic plant that carried genetically linked *pHTR10:cals3m* and *pRPS5A:H2B-tdTomato* (*SC-cal RHT* hemizygous plant identical to the line used in Figs. 2 and 3). According to the number of pollen tubes (PT, arrows) entered and the number of tdTomato-labeled nuclear foci (arrowheads) at the micropyle, 498 ovules from nine pistils were classified into seven patterns. Scale bar: 50 μm.

## Genetic dissections of vegetative nucleus transport and sperm cell transport.

The pollen-tube growth assays described above revealed the defective transport of sperm cells that accumulated callose. We measured sperm cell positions to characterize this phenotype. We generated three independent lines of the *SC-cal RHT* hemizygous plants and wild-type *RHT* hemizygous plants and then analyzed their pollen tubes with the tdTomato-labeled nuclei at 6 h after incubation on the growth medium. In wild-type *RHT* hemizygous plants, the vegetative nucleus positioned ahead of the two sperm cells was observed in the apical region in 96–100% of pollen tubes (VN&SN apical, VN ahead, Fig. 5a, f, g). In the *SC-cal RHT* hemizygous plants with strong genetic linkage

between *SC-cal* and *RHT*, sperm nuclei were retained in the basal region in 68–79% of pollen tubes (VN apical, SN basal, Fig. 5b, f). Indeed, the distance from the pollen-tube tip to the sperm nuclei significantly lengthened in the *SC-cal RHT* plants than in the wild-type *RHT* plants, but their vegetative nucleus positions were comparable (Fig. 5i, j). We concluded that callose expression in sperm cells inhibited transport of the sperm cells to the apical region of the pollen tube without adversely affecting the transport of the vegetative nucleus.

According to our results, the *SC-cal* line could be used to distinguish sperm cell transport from vegetative nucleus transport, which is specifically affected in the *wit1 wit2* double mutation (hereafter, *wit1/2*)[6]. To assay the interaction between these two genetic backgrounds, we used a *wit1/2* homozygous mutant[6]. We generated three independent *wit1/2* mutant plants hemizygous for the *RHT* nuclear marker and analyzed the tdTomato-positive nuclei in the pollen tube 6 h after germination. The vegetative nucleus remained in the basal part of 8–23% of the *wit1/2* mutant pollen tubes (Fig. 5d, f, SN apical, VN basal), which was in contrast with the sperm cell-immotile *SC-cal RHT* pollen tubes (Fig. 5b). As previously reported[6], most the other pollen tubes (77–90%) displayed apical transport of the unit of vegetative nucleus and sperm cells, while the orientation of the unit was inverted with the sperm nuclei first in 93–96% of the *wit1/2* pollen tubes (Fig. 5c, f, g, *wit1/2 RHT*, VN&SN apical, SN

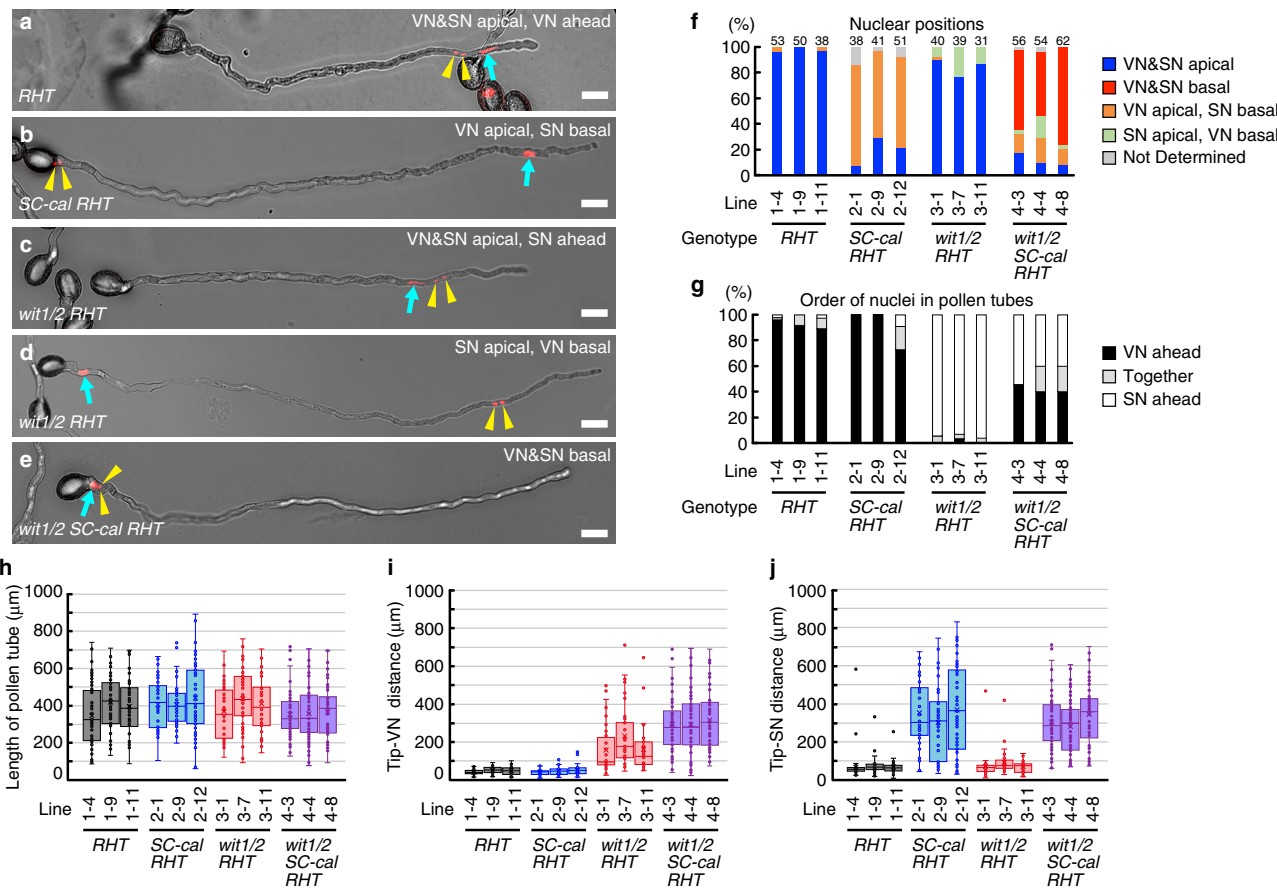

**Fig. 5 Genetic dissections of vegetative nucleus transport and sperm cell transport. a–e** Representative images of in vitro-germinated pollen tubes in wild-type and nuclear transport-defective plants. A transgenic line carrying only *pRPS5A:H2B-tdTomato* nuclear marker (*RHT*, **a**). A transgenic line carrying the *pHTR10:cals3m* and *RHT* (*SC-cal RHT* hemizygous plants, *SC-cal RHT*, **b**). A *wit1 wit2* double homozygous mutant carrying the *RHT* (*wit1/2 RHT*, **c**, **d**). A *wit1 wit2* double homozygous mutant carrying the *SC-cal* and *RHT* (*wit1/2 SC-cal RHT*, **e**). Cyan arrows and yellow arrowheads indicate the vegetative nuclei (VN) and pairs of sperm nuclei (SN), respectively. Scale bar: 20 μm. Representative images obtained using more than seven independent assays are shown. **f**, **g** Nuclear positions in pollen tubes (**f**) and nuclear orders (**g**) analyzed in three independent lines of *RHT*, *SC-cal RHT*, *wit1/2 RHT*, and *wit1/2 SC-cal RHT*. **h–j** Box-and-whisker plots of pollen-tube length (**h**), distance from pollen-tube tip to the vegetative nucleus (**i**), and distance from the pollen-tube tip to the sperm nucleus (**j**), analyzed in the same samples in (**f**, **g**). Numbers of pollen tubes analyzed in (**f–j**) are shown above the bars in (**f**). The values of each sample in (**f–j**) are obtained using more than seven independent experiments. Box-and-whisker plots show median (center line), mean (cross mark), upper and lower quartiles (box), maximum and minimum (whiskers), and points and outliers (solid circles). Note that only the pollen tubes with the *RHT* marker were analyzed and all the lines shown in this figure were independent from those shown in Figs. 1–4.

ahead). Although the vegetative nucleus was apparently distant from the pollen-tube tip, the positions of sperm nuclei were similar to those in the wild-type (Fig. 5i, j), highlighting the specificity of the apical transport defect in the vegetative nucleus of the *wit1/2* pollen tubes.

**Generation of a physiologically enucleated pollen tube**. In *wit1/2* double mutants hemizygous for the *SC-cal* and *RHT* with strong genetic linkage (hereafter, *wit1/2 SC-cal RHT* hemizygous plants), we observed that the vegetative nucleus and sperm cells were isolated in the basal region in 50–76% of the tdTomato-positive pollen tubes 6 h after germination (Fig. 5e, f *wit1/2 SC-cal RHT*, VN&SN basal). Time-lapse movies of germinating pollen tubes from the *wit1/2 SC-cal RHT* hemizygous plants showed that the sperm nuclei actively moved in the pollen grain (Supplementary Movie 3), indicating that the transport defect of the *wit1/2* mutant vegetative nucleus enhanced by *SC-cal* was likely due to the loss of pulling force by sperm cells rather than tethering by sperm cells inside the pollen grain.

As shown in Fig. 5e, the immotile sperm nuclei and vegetative nucleus were located behind the first callose plug. The Columbia-

0 (Col-0) ecotype used in this study formed the first callose plug, approximately 50 μm of the pollen grain[26]. To examine callose-plug formation in the *wit1/2 SC-cal RHT* hemizygous plants, we analyzed a time course of pollen-tube germination. The *wit1/2 SC-cal RHT* pollen tubes containing three tdTomato-labeled immotile nuclei generated the first callose plug within 3 h after germination, which was similar in Col-0 wild-type pollen tubes (Supplementary Fig. 4a–i). The number of callose plugs in a single pollen tube was also similar in the *wit1/2 SC-cal RHT* hemizygous plants to that in the Col-0 wild-type (Supplementary Fig. 4j), suggesting that the deposition of callose plugs is independent of the positions of the sperm cells and vegetative nucleus.

The isolation of the sperm nuclei and vegetative nucleus at the basal region may terminate nascent transcript supply to the apical region that contains no nucleus. To investigate this possibility, we observed cross-sections around the first callose plug in pollen tubes containing three tdTomato-labeled immotile nuclei in *wit1/2 SC-cal RHT* by transmission electron microscopy (Supplementary Fig. 5). A cross-section of the non-callose-plug area had a large vacuole (Supplementary Fig. 5b). In contrast, another section exhibited the congregation of

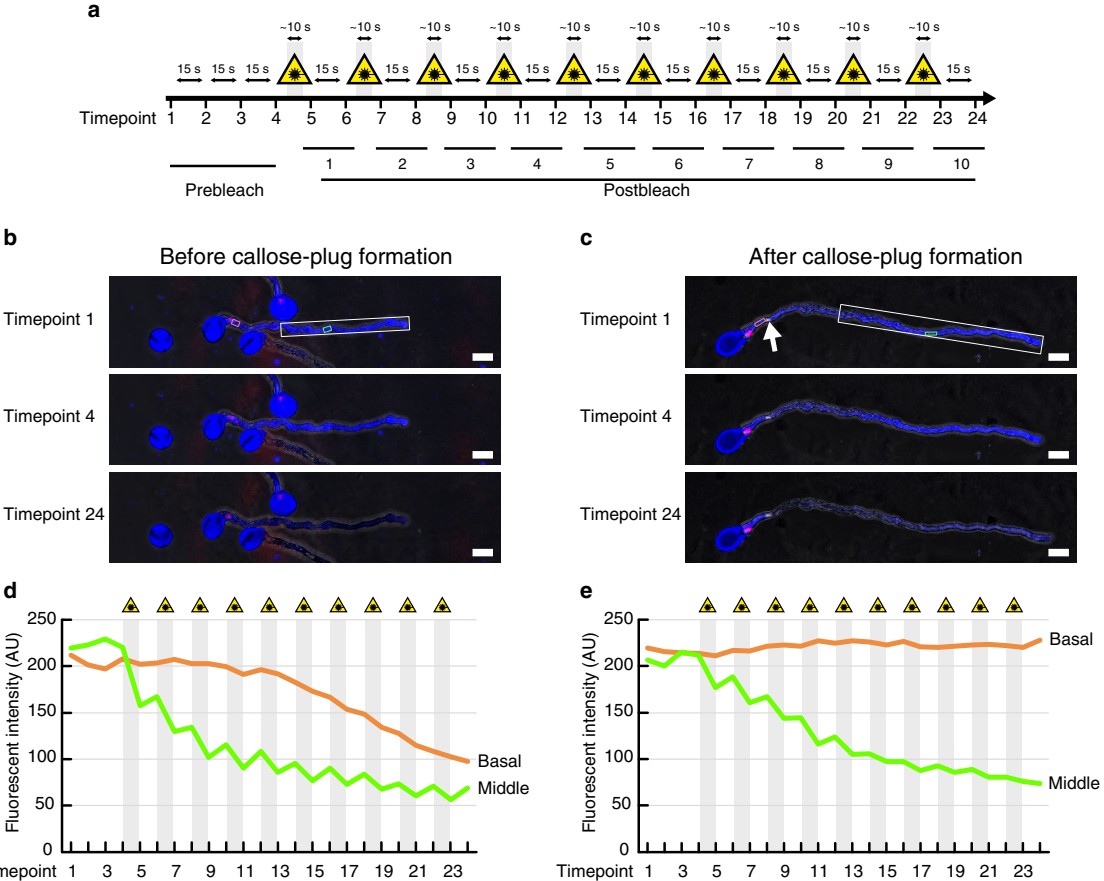

**Fig. 6 Cytoplasm isolation after the first callose-plug formation. a** Experimental flow of photobleaching. Pollen tubes on a growth medium were subjected to 10 rounds of photobleaching with 405 nm laser irradiations (yellow triangles and gray windows) 2 h after germination. **b**, **c** Pre- or post-bleached pollen tubes from a *wit1 wit2* double mutant hemizygous for *pHTR10:cals3m* (*SC-cal*), *pRPS5A:H2B-tdTomato* (*RHT*), and *pLAT52:mTurquoise2*, a reporter gene of cytosol in the vegetative cell. Both images are representative of three pollen tubes showing similar patterns. *SC-cal* and *RHT* were genetically linked. Pollen tubes containing tdTomato-labeled basal sperm cells and vegetative nucleus were analyzed before (**b**) or after (**c**) callose-plug formation. White boxes represent photobleached area. White arrow indicates the callose plug. Scale bar: 20 μm. **d** Fluorescent intensity of the photobleaching in (**b**). **e** Fluorescent intensity of the photobleaching in (**c**). The areas indicated by orange or green boxes at Timepoint 1 in (**b**) or (**c**) were analyzed in (**d**) or (**e**), respectively.

vesicles containing non-membranous material with medium electron density that resembled callose grains in the *Petunia* pollen tube[27], which likely corresponds to the growing region of the first callose plug (Supplementary Fig. 5c). Furthermore, we found this non-membranous material inside the pollen tube in other sections (Supplementary Fig. 5d, *n* = 3), suggesting the integrity of the first callose plug is completed.

To further examine the callose-plug integrity, we performed a fluorescence loss in photobleaching (FLIP) assay using a *wit1/2 SC-cal RHT* hemizygous plant that was also hemizygous for a vegetative-cell-specific cytosolic marker, *pLAT52:mTurquoise2* (Fig. 6 and Supplementary Movie 4). At 3 h after germination on the growth medium, mTurquoise2 was distributed in the enucleated apical region and in the basal region containing immotile sperm cells and a vegetative nucleus (Prebleach, Timepoint 1–4 in Fig. 6a–c). Photobleaching by 10 rounds of 405 nm laser irradiation rapidly reduced the mTurquoise2 signal in the apical half of the pollen tubes (Postbleach at Timepoint 24 in Fig. 6a–c; 'Middle' in Fig. 6d, e). Accompanied by the signal change in the apical region, the mTurquoise2 signal also decreased in the basal area of the pollen tube before the formation of the first callose-plug (*n* = 3, Fig. 6b; 'Basal' in Fig. 6d). However, we did not observe the signal reduction in the basal area behind the callose plug after the formation of the first

callose-plug (*n* = 3, Fig. 6c; 'Basal' in Fig. 6e). This indicates that the presence of the callose plug is preventing the movement of the mTurquoise2 protein from the basal regions to the growing regions of the pollen tube, and that potentially the movement of other molecules is also prevented. Taken together, the apical portion of the pollen tube may appear in a physiologically enucleated condition following callose-plug formation, since the callose plug likely terminates the de novo transcript supplied from the immotile vegetative nucleus in the basal region.

**Growth and guidance of physiologically enucleated pollen tubes.** The *wit1/2 SC-cal RHT* hemizygous plants showed a variety of defects in the positions of the sperm cells and vegetative nucleus in the pollen tubes (Fig. 5f). Nevertheless, the length of the pollen tubes in *wit1/2 SC-cal RHT* hemizygous plants was similar to that in the wild-type, implying that the retention and isolation of the vegetative nucleus had a limited effect on pollen-tube growth (Fig. 5h). The resilience of enucleated pollen tubes may be sustained by transcripts or proteins synthesized before callose-plug formation. To analyze protein stability, we used the *pLAT52:mTurquoise2* and the *pPRK6:PRK6-mRUBY2*, two reporter genes expressed in mature pollen and growing pollen tube[28,29]. In pollen tubes from *wit1/2 SC-cal RHT* hemizygous

plants carrying *pLAT52:mTurquoise2*, we observed cytosolic mTurquoise2 in the apical region even 8 h after the germination (Fig. 7a). We then analyzed pollen tubes from the *wit1/2 SC-cal RHT* hemizygous plants harboring the *pPRK6:PRK6-mRUBY2* (Fig. 7b). PRK6 is a receptor-like kinase that perceives pollen-tube attractant AtLURE peptides[28]. In the enucleated pollen tubes, the signal of tip-localized PRK6-mRUBY2 was detected in the same level as that in the tdTomato-negative control pollen tubes segregated from the same plant (Fig. 7c).

According to the stability of PRK6-mRUBY2, the enucleated pollen tubes may retain the capability to respond to pollen-tube attractant signals. Thus, we performed a modified semi-in vitro pollen-tube attraction assay to investigate the ovular-targeting behavior of a single pollen tube (Fig. 8a)[30]. In each assay, a pollen grain was attached to a papilla cell of a dissected pistil on pollen-tube growth medium and incubated with wild-type ovules for more than 14 h, followed by aniline blue staining to analyze the growth patterns of the pollen tubes. We validated the single pollen-tube guidance assay by using ovules from wild-type Col-0 and *myb98* homozygous mutant defective in the attraction of pollen tubes[31]. We confirmed that 20.4% of wild-type pollen tubes entered Col-0 ovules ($n = 54$, Supplementary Fig. 6a, c, d), but none penetrated *myb98* ovules ($n = 54$, Supplementary Fig. 6b–d). When we analyzed the wild-type Col-0 pollen, 80% of the pollen tubes penetrated the cut style ($n = 115$, Fig. 8b). The *wit1/2* double mutant, which was also hemizygous for the *SC-cal RHT*, produced tdTomato-positive *wit1/2 SC-cal* pollen and tdTomato-negative *wit1/2* pollen. After assessing the tdTomato signal under a fluorescent microscope, each pollen grain was subjected to a single pollen-tube attraction assay. Pollen tubes that emerged through the cut style were observed in 57.5% of pistils pollinated with the tdTomato-positive *wit1/2 SC-cal* pollen ($n = 134$). This frequency did not significantly differ from those of wild-type Col-0 pollen or tdTomato-negative *wit1/2* pollen (54.8%, $n = 104$, Fig. 8c, e), confirming the growth capability of the tdTomato-positive *wit1/2 SC-cal* pollen tubes (Fig. 5e, h). Surprisingly, the insertion of the pollen tube into the ovule occurred with 28.6% of tdTomato-positive *wit1/2 SC-cal* pollen (Fig. 8d, f, *wit1/2 SC-cal + RHT +*), which was comparable to that in Col-0 pollen (23.9%, Fig. 8b, f, Col-0) and tdTomato-negative *wit1/2* pollen (42.1%, Fig. 8c, f, *wit1/2 SC-cal − RHT −*). Among the 22 *wit1/2 SC-cal RHT* pollen tubes displaying successful attraction, 73% were found to have the unit of vegetative nucleus and sperm nuclei confined to the basal area around the papilla cells (Fig. 8d, right panels). These data revealed a previously hidden pollen-tube ability to grow and respond to the female signal even in the absence of the vegetative nucleus and sperm cells at their tip.

## Discussion

The features of the pollen tube, including active tip growth and response to female tissues, were thought to be controlled by gene expression from the vegetative nucleus[11]. By inducing sperm cell-specific callose accumulation in the *wit1/2* double mutant, we generated pollen tubes whose vegetative nucleus and sperm cells were isolated in the basal region by callose-plug formation. This aberrant apical nuclear transport revealed the unexpected independence of the directional growth control of the pollen tube from the tip-localization of the vegetative nucleus.

We found that the dominant mutant of *CALLOSE SYNTHASE 3* expressed from a sperm cell-specific *HTR10* promoter (*SC-cal*) could induce defects in the apical transport of sperm cells (Fig. 3). Independent motive forces of the vegetative nucleus and sperm cell have been discussed in previous studies[32,33]. This idea was supported by another previous report that observed the

compromised apical transport of the vegetative nucleus in the *wit1/2* double mutant[6]. *SC-cal*-induced loss of sperm cell motility is important evidence for determining the independence of vegetative nucleus transport and sperm cell transport.

It is unclear why *SC-cal* impaired sperm cell transport. In *A. thaliana*, heat shock stress also results in the retention of sperm cells in the basal area of the pollen tube[34]. Since heat-induced transport defects are accompanied by the separation of a pair of sperm cells, the dissociation of the inner vegetative plasma membrane enclosing the sperm cells might be involved in the loss of sperm cell motility. Although Lyn24-mNeonGreen was less frequently localized at the inner vegetative plasma membrane in the *SC-cal* pollen (Fig. 2e–g), sperm cells were usually observed as pairs (Fig. 3e, g, i and Supplementary Movies 1–3). Therefore, callose accumulation on the sperm cells may affect their transport via alterations in the lipid or protein composition of the inner vegetative plasma membrane, rather than the disappearance of the inner vegetative plasma membrane. In contrast to animals and non-flowering plants, the sperm cells of flowering plants do not have structures generating motive force such as flagella or pilli[1]. Hence, the mechanism of sperm cell migration in pollen tubes has been a long-standing mystery of plant reproduction. Further biochemical and genetic approaches would provide answers to remaining questions regarding the relevance of callose levels, alteration of the inner vegetative plasma membrane, and motility of sperm cells.

The retention of the unit of vegetative nucleus and sperm cells and subsequent callose-plug formation would likely interrupt nascent transcript supply to the apical region (Fig. 6), but the growth and ovular-targeting behavior of the pollen tubes were normal (Fig. 5h and Fig. 8f). Many plants, including *A. thaliana*, show tolerance against transcription inhibitors during germination and tip growth, but the effect varies among species[7,35], suggesting that the early stages of pollen tubes use mRNA accumulated before pollen germination. However, transcriptome or translatome analyses have identified subsets of genes expressed after germination[7–10]. Indeed, some of these genes have been shown to play important roles in the late stage of pollen tubes, such as ovular-targeting behavior[9,10]. Most likely, even the late-stage pollen-tube functions can be sufficiently maintained by transcripts expressed for a few hours after germination until callose-plug formation. Conversely, not all pollen-tube functions might be sustained by pre-set gene expression. Previous studies reported that the *wit1/2* double mutant showed defects in discharging pollen-tube contents into synergid cells; nevertheless, the extent to which the pollen-tube reception relies on the apically localized vegetative nucleus remains to be shown owing to the passive migration of the vegetative nucleus by motile sperm cells[6,36]. In the present study, we did not conduct a quantitative analysis of pollen-tube reception. However, the *SC-cal wit1/2* pollen tube can provide an ideal tool for answering this question and will provide new insights into the sophisticated delivery system of the male genome in flowering plants.

## Methods

**Gene identifiers**. The AGI codes of the genes in the present study are as follows: *WIT1* (AT5G11390), *WIT2* (AT1G68910), *CALS3/GSL12* (AT5G13000), *HISTONE 2B* (AT1G07790), *GCS1/HAPLESS2* (AT4G11720), *RANGAP1*(AT3G63130), *HTR10/ MGH3* (AT1G19890), *SYP132* (AT5G08080), *PRK6* (AT5G20690), and *ACA3* (AT5G04180).

**Plant materials and growth conditions**. *Arabidopsis thaliana* Columbia-0 (Col-0) was used as the wild-type plant and the background of all mutants. The *wit1-1 wit2-1* double null mutant was kindly provided by Dr. Meier and Dr. Tamura[6]. Plants were grown on soil at 22 °C under continuous light or standard long-day conditions (16 h light: 8 h dark).

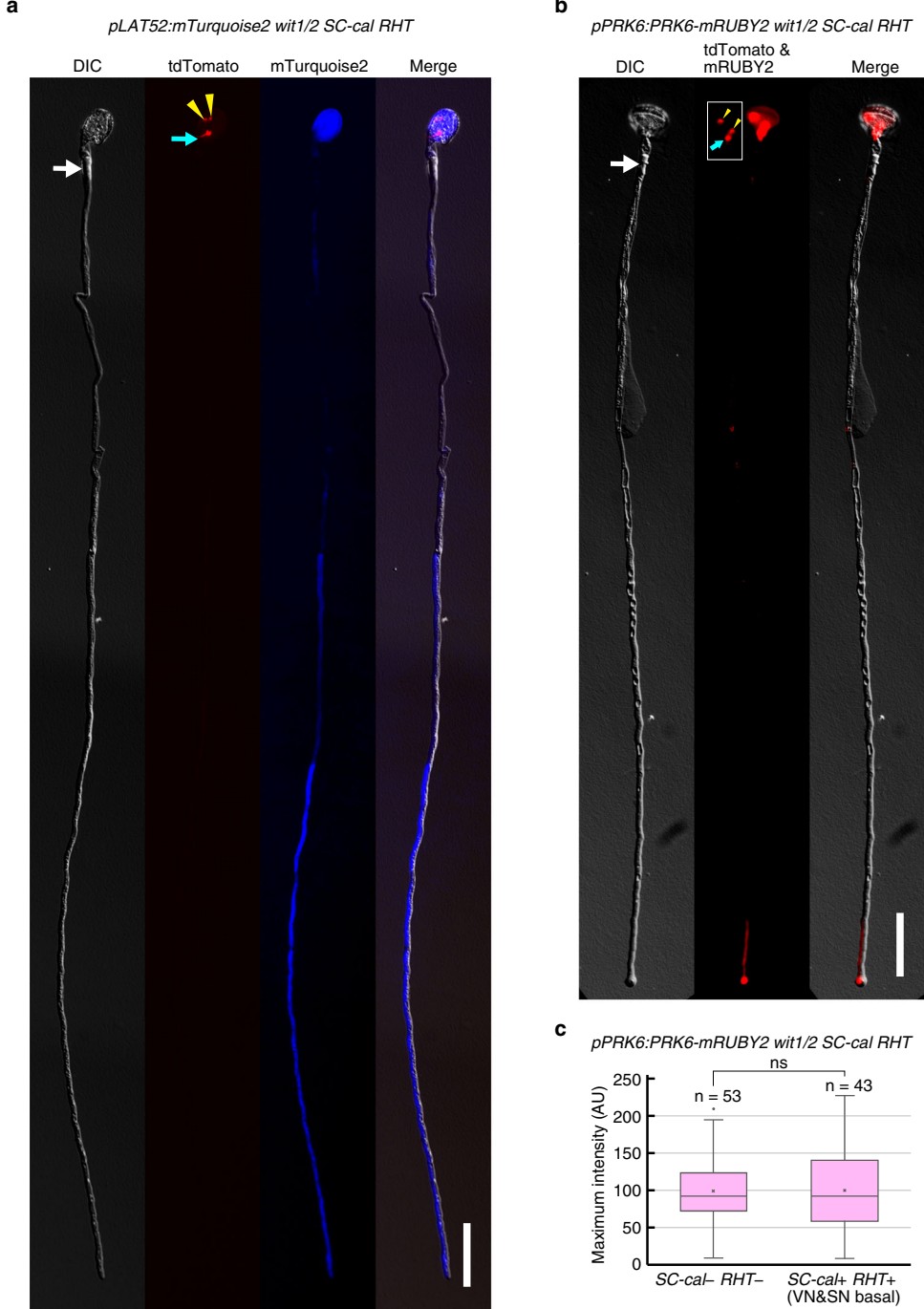

**Fig. 7 Detection of vegetative cell-specific reporter proteins in anuclear apical region. a, b** Pollen tubes expressing mTurquoise2 (**a**) or PRK6-mRUBY2 (**b**) that contained immotile sperm cells and vegetative nucleus in the basal area of those pollen tubes. Pollen from the *wit1 wit2* double mutant hemizygous for genetically linked *pHTR10:cals3m* (*SC-cal*) and *pRPS5A:H2B-tdTomato* (RHT), also hemizygous for the *pLAT52:mTurquoise2* (**a**) or *pPRK6:PRK6-mRUBY2* (**b**), were incubated on pollen-tube growth medium. To observe the long pollen tube in a single focal plane, pollen tubes containing immotile nuclear triplet were dragged out and transferred to fresh medium using a needle under a fluorescence stereomicroscope 8 h after germination and then observed under an epifluorescence microscope. Representatives of 20 and 15 pollen tubes with similar patterns are shown in (**a**) and (**b**), respectively. Inset in (**b**) represents tdTomato-labeled sperm nuclei and vegetative nucleus captured at shorter exposure time (3 s). **c** Box-and-whisker plot of PRK6-mRUBY2 signal in pollen tubes from the *pPRK6:PRK6-mRUBY2 wit1/2 SC-cal RHT* plant that was also shown in (**b**). At 8 h after germination, the maximum intensity of the red fluorescence was analyzed at the tip of pollen tubes expressing PRK6-mRUBY2. Box-and-whisker plots show median (center line), mean (cross mark), upper and lower quartiles (box), maximum and minimum (whiskers), and outlier (circle). A two-tailed Student's *t*-test indicated no significant difference between the anuclear pollen tube that left three tdTomato-positive nuclei in the basal region (*SC-cal* +, *RHT* + VN&SN basal) and the tdTomato-negative control pollen tubes (*SC-cal* −, *RHT* −) (*P* = 0.773). Yellow arrowheads, cyan arrows, and white arrows represent the sperm nuclei, vegetative nuclei, and first callose plugs, respectively. Scale bar: 50 μm.

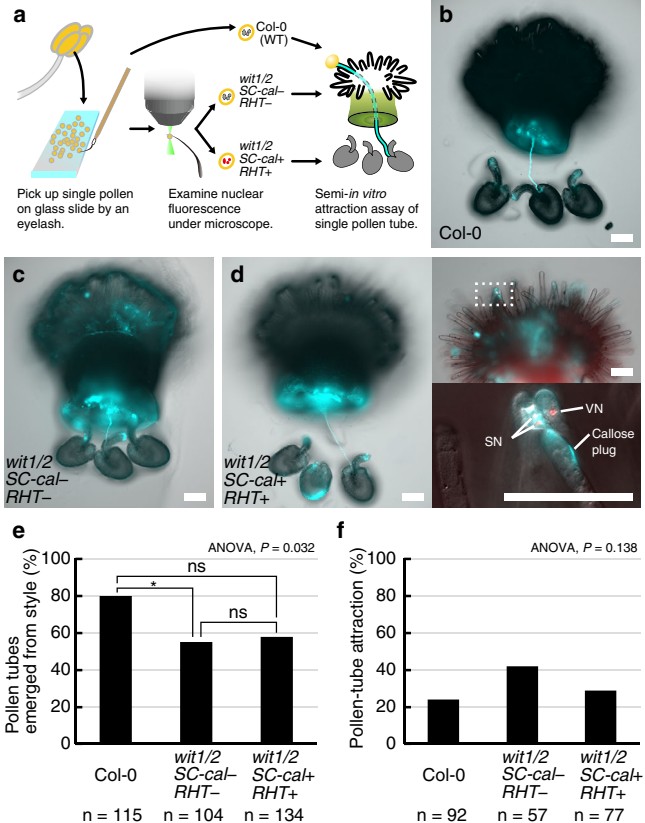

**Fig. 8 Guidance of pollen tubes containing immotile vegetative nucleus and sperm cells.** **a** Schematic drawing of the single pollen-tube guidance assay. **b–d** Representative images of ovular-targeting pollen tubes visualized by aniline blue staining after the single pollen-tube guidance assay. Pollen tubes from wild-type Col-0 pollen (Col-0, **b**). Pollen tubes from tdTomato-negative pollen (*wit1/2 SC-cal − RHT −*, **c**) or tdTomato-positive pollen (*wit1/2 SC-cal + RHT +*, **d**) from a *wit1 wit2* mutant line hemizygous for genetically linked *pHTR10:cals3m* and *pRPS5A:H2B-tdTomato* (identical to the *wit1/2 SC-cal RHT* line shown in Fig. 5). The right panels in (**d**) represent the *RHT +* pollen tube on the stigma; magnification of the dashed box in the upper-right is shown in the lower-right. Scale bar: 100 μm. Images are representative of 22 (**b**), 24 (**c**), or 22 (**d**) independent assays showing similar patterns. **e** Percentage of pollen tubes that emerged from the cut style. **f** Percentages of pollen tubes that entered the micropyle in the ovules. Statistical test: Bonferroni-adjusted significance test was performed after one-way ANOVA. Asterisk indicates significance ($P = 0.0438$).

**Plasmids.** A DNA fragment of the *HTR10* promoter (1215 bp upstream region from the ATG of the *HTR10* gene) was amplified from Col-0 genomic DNA using the primers 'pHTR10HindIII_F' and 'pHTR10HindIII_R' (Supplementary Table 1), and cloned into the *Hin*dIII site of the pGWB501 destination vector to produce pDM252[18,37]. pOR084, an entry clone containing mRUBY2-SYP132, was constructed as follows: (1) A pENTR/D-TOPO vector (Invitrogen) containing the *mRUBY2* gene was amplified using the primers 'FP_SpeI_F' and 'FP_EcoRI_R' (Supplementary Table 1);[38] (2) the coding region of the *SYP132* was amplified from a cDNA of a Col-0 seedling using the primers 'SYP132_EcoRI_F' and 'SYP132_-SpeI_R' (Supplementary Table 1); and (3) these two PCR products were assembled using the NEBuilder HiFi DNA Assembly Cloning Kit (New England Biolabs, MA, USA) to produce pOR084. LR recombination between pDM252 and pOR084 was performed using LR Clonase II (Invitrogen) to produce pDM439, a vector containing *pHTR10:mRUBY2-SYP132*. Another LR recombination between pDM252 and pENTR cals3m gave rise to MNv231, a plasmid carrying the *pHTR10:cals3m* (*SC-cal*). A DNA fragment of the *ACA3* promoter[22] (1607 bp upstream region from the ATG of the *ACA3* gene) was amplified from Col-0 genomic DNA using the primers 'pACA3HindIII_F' and 'pACA3HindIII_R' and cloned into the *Hin*dIII site of the pGWB501 destination vector to produce pDM527 (Supplementary Table 1)[37]. A DNA fragment of *mNeonGreen* was amplified using primers 'Lyn24-

mNG_F' and 'FP_R' and cloned into pENTR/D-TOPO to produce pOR158 (Supplementary Table 1)[39]. LR recombination between pDM527 and pOR158 was performed to produce pDM549, a vector containing *pACA3:Lyn24-mNeonGreen*. dHT1, a plasmid carrying the *pRANGAP1:RANGAP1-mNeonGreen*, was constructed as follows: (1) *mNeonGreen* was amplified using primers 'Linker1_FP_F' and 'mNG_R' (Supplementary Table 1);[39] (2) the genomic sequence of the *RAN-GAP1* (1214 bp upstream of the start codon to the end of the protein-coding region without the stop codon of AT3G63130) was amplified using primers 'RAN-GAP1pro_F' and 'RANGAP1_R' (Supplementary Table 1);[6] and (3) these two DNA fragments were mixed with the plasmid backbone of HTv896, a pPZP211[40]-based plasmid containing the *NOS* terminator after the AviTag™ sequence (Avidity), and assembled using the NEBuilder HiFi DNA Assembly Cloning Kit to generate dHT1. The *LAT52* promoter (593 bp) originated from LAT52-7[29] was amplified using primers 'pLAT52HindIII_F' and 'pLAT52HindIII_R' (Supplementary Table 1), and cloned into the *Hin*dIII site of the pGWB501 destination vector to produce pDM249[37]. A DNA fragment of *mTurquoise2*[41] was amplified using primers 'pENTR_CFP_F' and 'FP_R' and cloned into pENTR/D-TOPO to produce pOR120 (Supplementary Table 1). LR recombination between pDM249 and pOR120 was performed to produce pDM473, a vector containing *pLAT52: mTurquoise2*. A pPZP221[40] plasmid containing *pPRK:PRK6-mRUBY2* was described previously[28]. *pGCS1:GCS1-Clover* was provided by Dr. Tomokazu Kawashima, Kentucky University. *pRPS5A:H2B-tdTomato* (RHT) was provided by Dr. Daisuke Kurihara[19].

**Preparation of transgenic plants.** Agrobacterium-mediated plant transformation was performed by the floral dipping method using Agrobacterium strain GV3101[42]. The simultaneous transformation of *SC-cal* and *RHT* was conducted as follows: (1) Agrobacterium suspensions of *SC-cal* and *RHT* were independently prepared and mixed just before the floral dipping procedure; (2) transformants harboring both *SC-cal* and *RHT* were selected on Murashige and Skoog medium containing 50 μg/mL hygromycin B and 50 μg/mL kanamycin; (3) screening of the *SC-cal RHT* transgenic lines was performed by fluorescent observation of mature pollen mounted in aniline blue solution (0.1% [w/v] aniline blue, 0.1 M K₃PO₄); and (4) to select transgenic lines with a strong genetic linkage between *SC-cal* and *RHT*, we confirmed the 1:1 segregation of the tdTomato-positive and -negative pollen and then determined whether the presence or absence of the tdTomato signal corresponded to the aniline blue signals on the sperm cells in more than 90% of those pollen.

**Analysis of pollen grains.** In the analysis of callose deposition during microsporogenesis, stamens in various flowering stages were mounted in the aniline blue solution (see the 'Methods' section on 'Preparation of transgenic plants') and squashed to release pollen from the anthers. Fluorescent images of tdTomato and aniline blue were photographed using an Olympus IX73 inverted microscope (Tokyo, Japan) equipped with an sCMOS camera (Zyla 4.2; Andor, Belfast, UK) using RFP and CFP filters, respectively. To obtain a super-resolution image, we used the deconvolution software Huygens Ver. 17.04. (Scientific Volume Imaging, Hilversum, the Netherlands) from an image acquired using the photon-counting mode of a confocal laser scanning microscope (Leica TCS SP8, Wetzlar, Germany). Pollen from *SC-cal RHT* carrying the *pGCS1:GCS1-Clover* or *pACA3:Lyn24-mNeonGreen* reporter gene were mounted in 5% sucrose and observed by the TCS SP8.

**Staining of ovules or seeds.** Congo red staining was performed as follows: (1) Pistils one-day-after-pollination were dissected and covered by a coverslip with 5% sucrose; (2) through the narrow space between the glass slide and cover slip, the samples were stained with 0.4% Congo red solution for 1 min and rinsed several times with 5% sucrose; (3) fluorescent signals were observed by the RFP filter set using the Olympus IX73 inverted microscope. In vanillin staining, seeds at 2 days post pollination were treated with a 6 N HCl solution containing 1% vanillin (w/v) for more than 1 h at room temperature[43].

**In vitro and semi-in vitro pollen-tube growth.** To observe the pollen tubes in vitro, pollen grains were incubated on a pollen germination medium (0.01% boric acid, 5 mM CaCl₂, 5 mM KCl, 1 mM MgSO₄, 10% sucrose, adjusted pH to 7.5 with 1 N KOH, 1.5% NuSieve GTG agarose supplemented with 10 μM epibrassinolide[44]) and observed by an Axio Imager A2 upright microscope (Zeiss, Jera, Germany) equipped with a cooled CCD camera (AxioCam HRm; Zeiss), by an LSM780-DUO-NLO laser scanning microscope (Zeiss), or by Axio Imager M1 upright microscope (Zeiss). For the semi-in vitro pollen-tube growth, we modified the protocol described by Susaki et al.[29]. In brief, pollen-tube growth medium (0.01% boric acid, 5 mM CaCl₂, 5 mM KCl, 1 mM MgSO₄, 10% sucrose, adjusted pH to 7.5 with 1 N KOH, 1.5% NuSieve GTG agarose) was melted at 65 °C and solidified on a glass bottom dish (D11130H, Matsunami Glass IND., LTD., Japan) to form a thin layer. Stigmas from emasculated wild-type pistils were cut with a needle and placed on the medium, followed by hand-pollination with pollen from transgenic plants. After 3–5 h of incubation, pollen tubes that emerged from the cut pistils were observed by the TCS SP8 microscope. To label with DNA in the sperm nuclei and vegetative nucleus, we used pollen tube

growth medium containing 2000-fold diluted SYBR Green I. Note that the SYBR Green I staining was only used in the semi-in vitro pollen-tube growth assay; SYBR Green I extensively labeled cytosol when pollen tubes were directly germinated on the SYBR Green I-containing medium. For time-lapse imaging, 0.5 h after pollination, pollen grains were placed on a pollen tube germination medium for observation. Z-series images were captured every 2 min using a spinning-disk confocal system (CSU-WI; Yokogawa Electric, Tokyo, Japan) with a 60× objective lens (UPLSAPO, NA.1.3; Olympus, Tokyo, Japan) and an electron-multiplying CCD camera (iXON3 888; Andor).

**Photobleaching assay.** Pollen was placed on a thin layer of growth medium in a ϕ35 mm petri dish and incubated for 2 h at 22 °C. The germinated pollen tubes were transferred to a large coverslip (24 mm × 40 mm, Matsunami) with the growth medium, sandwiched by another smaller coverslip, and gently squashed to observe whole pollen tubes in a single focal plane (18 × 18 mm, Matsunami). Then, viable pollen tubes displaying rapid protoplasmic streaming were subjected to 10 rounds of photobleaching by the SP8 TCS microscope (Leica). In each photobleaching procedure, the apical half of the pollen tube was scanned 10 times with a 405 nm diode laser at 100% laser power.

**Electron microscopy.** In vitro-germinated pollen tubes were observed 8 h after germination under the Olympus MVX10 fluorescence stereomicroscope (Olympus) using the RFP filter set. Pollen tubes whose vegetative nucleus and sperm nuclei remained in the basal area were dragged out one-by-one and aligned in one place on the medium with a 27 gauge needle. Those pollen tubes were collected with small pieces of growth medium and fixed with 2% glutaraldehyde and 2% paraformaldehyde in 50 mM cacodylate buffer pH 7.4 at 4 °C. The tissue segments were washed in buffer and post-fixed for 3 h in 2% aqueous osmium tetroxide at 4 °C. The tissue was then dehydrated in a graded ethanol series, transferred into propylene oxide, infiltrated, and embedded in Quetol 651. Cross-sections around the basal area of the pollen tubes (90 nm thickness) were stained with 2% aqueous uranyl acetate and lead citrate and examined using a JEOL JEM 1400Plus electron microscope at 100 kV.

**Single pollen-tube attraction assay.** Pollen grains were spread on glass slides using an eyelash attached to a toothpick. From the glass slides, single pollen grains were picked up by the eyelash and subjected to semi-in vitro pollen-tube growth as described above. In the case of the *wit1 wit2* mutant hemizygous for the *SC-cal*-linked *RHT*, the presence or absence of the tdTomato signal was assessed using an upright Axio M1 Imager microscope (Zeiss) before pollination. Three fresh ovules from the wild-type or *myb98* mutant were placed adjacent to the cut style and incubated overnight. After the aniline blue staining, pollen tube and pollen grains on the stigma were observed by the IX73 microscope (Olympus).

**Reporting summary.** Further information on research design is available in the Nature Research Reporting Summary linked to this article.

## Data availability

All data that support the findings of this study are available from the corresponding authors upon reasonable request. Source data are provided with this paper.

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

## Acknowledgements
We thank K. Tamura and I. Mayer for *wit1 wit2* mutant; Y. Helariutta and S. Miyashima for *cals3m* construct; D. Kurihara for *pRPS5A:H2B-tdTomato*; T. Kawashima for *pGCS1: GCS1-Clover*; K. Matsuura-Tokita for in vitro germination assay, M. Tsukatani, H. Ikeda, and H. Kakizaki for technical supports, F. Berger for advise on the research, Y. Nakaoka for image processing, Tokai Electron Microscopy, Inc. for transmission electron microscopy, and Editage for English language editing. This work was supported: by JSPS KAKENHI Grant Numbers (Nos. 16H06173, 17H05846, 19H04869, 20H03280, 20K21432, 20H05422, 20H05778, and 20H05781 to D.M., Nos. 16J02257, 19K23759, and 20K15822 to K.M., Nos. 16H06464, 16H06465, 16K21727, and JP16H06280 to T.H.); by JST PRESTO Grant Numbers (No. JPMJPR20D9 to K.M.); by Grant for academic research from Yokohama City University (to D.M.); by the grant for 2016–2021 Research Development Fund of Yokohama City University (to D.M.); by the grant for 2019–2021 Strategic Research Promotion (Nos. SK1903) of Yokohama City University (to D.M.); by the Sasakawa Scientific Research Grant from The Japan Science Society (to K.M.); and by the Third-Phase R-GIRO Research from the Ritsumeikan Global Innovation Research Organization, Ritsumeikan University (to K.M. and A.T.).

## Author contributions
M.N. and D.M. conceived the idea of gametophyte-specific *cals3m* expression system. K.M. and D.M. performed most of experiments, directed the project, and wrote the manuscript. H. Tsuchi constructed a plasmid of the *pRANGAP1:RANGAP1-mNeonGreen*. H. Takeuchi, A.T., T.K., and T.H. provided pivotal suggestions on the manuscript.

## Competing interests
The authors declare no competing interests.
