## [Peer Review File · Nature Communications]

REVIEWER COMMENTS

Reviewer #1 (Remarks to the Author):

The authors show in this manuscript an interesting study about pollen tube growth and the position of the nuclei. The main contribution of this study is to report a normal pollen tube growth but containing sperm and vegetative basal. It demonstrates that the apical position of the nuclei is dispensable for pollen tube growth. To observe this innovative result, the authors perform interesting experiments by analysing transgenic plants. The experiments have been conducted rigorously with enough replications and support the conclusion. The authors evaluate anomalous callose deposition around sperm cells by making the transgenic line SC-cal containing a nuclear marker. The double mutant *wit1wit2*, previously studied by Zhou and Meier 2014, has also been evaluated in this manuscript, and together with SC-cal, the authors have observed the unusual position of the nuclei, but maintaining normal pollen tube growth. This result is relevant for the study of the mechanisms involved in pollen tube growth and fertilization of flowering plants. I only have a few suggestions and minor points that would help the readability of the paper.

Line 101. It would be more accurate to say 'To reveal sperm cells.'

To further clarify the figure 2, the authors may add SC-cal - and SC-cal + in RHT - and RHT + respectively or the transgenic line carrying the RHT and *pHTRR10::cals3m* can also be named SC-cal, as in the Fig.5 (line 585). The indication of the schematic representation in Fig2a should be more accurate. The callose deposition is clear in the super-resolution image, but have not been included in the schematic representation of RHT+.

Line 124. Is this transgenic plant the same one that the transgenic line in the fig 2a? The authors should indicate this in the text for clarity, as the authors describe for the SC-cal RHT transgenic plant of Fig2a, Fig. 4 and Fig. S3 (line 667-668).

Reviewer #2 (Remarks to the Author):

This manuscript by Motomura and colleagues describes observations of a series of transgenic *Arabidopsis thaliana* lines to explore the localization of nuclei in growing pollen tubes. Based on their observations, the authors conclude that growth and guidance of pollen tubes can occur independently of a tip-localised vegetative nucleus. This is a novel idea and differs from the general dogma in the area that transcription/translation in the vegetative cell during pollen-tube growth is necessary for continual tip growth and communication with the female tissues. These main conclusions and focus (title, abstract) are drawn from pollen without any nuclei in the pollen-tube tip, but most of the manuscripts focusses on pollen tubes with altered sperm-cell position with the data on the enucleate pollen being brief and a little preliminary.

The majority of the manuscript describes the impact of ectopic callose expression in sperm cells (SC-Cal). The authors conclude that the SC-Cal sperm cells are no longer tethered to the vegetative cells and are commonly not transported in the pollen-tube tip. The data generally supports these conclusions, and the SC-Cal plants could be a useful tool to study physical interaction between the sperm cells and the vegetative nucleus.

I do however have one note on this part of the work. The authors conclude that "ectopic callose deposition on sperm cells caused male sterility" (line 94) and refer to male sterility several times through the manuscript. The data presented does not support this with Fig. 1d indicating up to a 40% transmission efficiency of the transgene (assuming a hemizygous parent) through the male in reciprocal crosses. This shows reduced transmission, not sterility. The manuscript should be edited to reflect this, and the interpretation of the data in the following sections take this into account (e.g. Fig. 4).

The authors then combine the SC-Cal construct with the previously published *wit1/2* double mutant that impacts vegetative-cell movement into the pollen tube, with the authors confirming the *wit1/2* phenotype. In vitro grown pollen tubes from the SC-Cal *wit1/2* plants were analysed, but the description of the results appears inconsistent with the data presented, potentially as it is unclear what was analysed (all pollen from a *wit1/2* plant hemizygous for SC-cal or only the pollen with the RHT marker linked to SC-cal?). Line 193 of the text states "In *wit1/2* double mutant plants hemizygous for the SC-cal and RHT with strong genetic linkage, we observed that vegetative nucleus and sperm cells were isolated in the basal region and the apical region contained no nucleus in 85-94% of the pollen tubes (Fig. 5e,f SC-cal *wit1/2*, VN&SN basal)". But this differs from the data in Fig. 5f where the red bars labelled as VN&SN basal look to be ~50 to 75% of the total. Additionally, in Fig. 5f it appears between 20 to 30% of the SC-cal *wit1/2* pollen tubes had an apical vegetative nucleus (VN&SN apical and VN apical, SN basal combined), thus suggesting a substantial amount of the SC-cal *wit1/2* pollen-tube tips do have at least one nucleus. As there is a range of phenotypes in the SC-cal *wit1/2* pollen, it needs to be clearer what was analysed and caution needs to be taken when analysing data from these plants as a population (Fig. 5 h to j).

The authors then describe an elegant system to test the growth and guidance of a single pollen tube with a known genotype. This provides some evidence that enucleate pollen tubes can be guided to an ovule, with the authors reporting guidance of pollen tubes with an observed basal vegetative nucleus and sperm cells. However, considering that this is the key conclusion of the paper and there needs to be confidence in this data, but I have a few concerns. Why are the RHT+ pollen tubes compared only to RHT- pollen tubes in Fig. 6e and only to WT pollen tubes in Fig. 6f? The authors use this data to conclude there is no impact of pollen-tube growth (6e) and guidance (6f) in RHT+ plants, but the opposite interpretation could be made if the genotype being compared were reversed, i.e. reduced guidance of RHT+ pollen tubes (~29%) compared to RHT- pollen tubes (~41%). I also wonder about the reliability of the assay, and whether placement of the ovules is such that a certain number of pollen grains will grow to the ovule. It would therefore be good to know how the guidance of RHT+ tubes compares to tubes with impaired guidance (a guidance mutant), if the results are statistically significant and how reproducible the assay is.

The idea that the vegetative nucleus, and notably transcripts from the vegetative nucleus, are not required for normal pollen-tube growth and guidance to ovules differs from previous reports, as discussed briefly in the conclusion. The authors speculate that this is because there is sufficient transcript abundance before the nuclei are sealed off from the pollen-tube tip by a callose plug. It would be great if there was some experimental evidence to support this. While RNA localisation is difficult, such an experiment could involve a fluorescent marker for a pollen-tube expressed gene to confirm the protein is still present (and quantitate the amount) in a pollen-tube tip with a basally-located vegetative nucleus and sperm cells, or RT-qPCR analyses on pollen tubes after emergence for a cut style from the different genotypes. While these experiments may not be necessary, they would add depth to the manuscript.

Minor corrections

For Figure 4 the legend states that 48 ovules from nine pistils were classified into seven patterns. Representative images from the nine classes are shown, but it would be best to also show the numbers in each class.

Be consistent with nomenclature around *Arabidopsis* or *Arabidopsis thaliana* (currently a mix).

Please provide AGIs for all genes used.

Genetic constructs are generally written as promoter: coding region (rather than promoter::coding region).

When "pollen tube" is being used as a descriptor it should be hyphenated (e.g. pollen-tube growth, pollen-tube function).

Abstract Line 26: change to "In the pollen tube, the vegetative...".

Line 90: change to "hygromycin-resistance in the F1...".

Line 509: Figure 1 legend change "in the sperm cell" to "in sperm cells".

Line 512 Add "plants" to "Aniline blue staining of mature pollen from wild-type (WT, a) or hemizygous pHTR10::cals3m (SC-cal, b) plants."

Line 513 should be "Transmission of SC-cals ...".

Relabel the y axes Fig 6e to Pollen tubes emerged from style.

Line 221: Remove the s from "Discussions".

Please provide a reference for the sentences on lines 222 to 224 and lines 256-257.

References: The titles of journal articles are in a mix of title case and sentence case.

Reviewer #3 (Remarks to the Author):

In this exciting manuscript, the authors challenge the long-standing assumption that the positioning of the vegetative nucleus at the tip of the pollen tube is necessary for pollen tube function. They use some clever genetic tools to generate mutant pollen tubes whose sperm cells and vegetative nucleus stay near the pollen grain and show that these pollen tubes can still grow and respond to attractant signals emitted from the ovules. The data are of generally high quality and the results will be of great interest to the plant reproduction community and scientists interested in cell biology, nuclear movement, and intercellular communication.

The main takeaway message from the paper is that the vegetative nucleus and sperm cells don't need to be at the tip for the pollen tube to be able to find ovules. The authors try to take the conclusions one step further with the "physiologically anuclear" concept. I think that this description/terminology is based on an assumption that callose plugs are normal and that the tip of the pollen tube has cytoplasm that is isolated from the nuclei that are stuck at the top of the pollen tube. The authors should consider the possibility that apical positioning of the male germ unit is required for normal callose plug formation as the pollen tube grows. If the mutants have abnormal callose plugs, then the pollen tube could be a big continuous cell even when the male germ unit is not at the tip. The authors should investigate these pollen tubes more if they want to claim that the pollen tubes are physiologically anuclear. Some additional experiments to quantify callose plugs (number and spacing) and cytoplasmic streaming in the mutants would provide evidence that these pollen tubes are behaving as anucleate cells.

The single pollen tube experiments presented in Figure 6 clearly show that the mutant pollen tubes with the male germ unit stuck near the pollen grain can still be targeted to ovules. It is interesting that the SC-cal seems to act as an enhancer of the *wit1/2* phenotype since the main phenotype of *wit1/2* mutants is reversed order of sperm and vegetative nuclei and the *wit1/2* mutants only rarely have a basal VN unless combined with SC-cal. Does this suggest that the sperm cells actually "hold" the VN in place and prevent its movement in this triple mutant? It is also interesting that the sperm cells tend to separate from the VN in SC-cal, but not in the triple mutant (unless they are actually not connected when they are all basal). The authors should consider adding some more discussion on the influence of the sperm on VN movement when they discuss the sperm having their own motive force.

Minor points

1. Line 79: The rationale for expressing *cals3m* in sperm cells might be difficult for non-experts to understand. The concept that sperm cells don't normally have a cell wall and that pollen tubes have a lot of callose in their cell walls and make callose plugs should be included in the introduction or in the leadup to this experiment.
2. The transmission analysis in Fig 1d suggests that SC-cal can be transmitted, but only hemizygous mutants are used in the experiments throughout the paper. The authors should comment on why hemizygous mutants were used (i.e. is it possible to get homozygous progeny?)
3. The genetic linkage between SC-cal and RHT should be introduced the first time this genotype is used. The description "cosegregation of callose accumulation and nuclear *tdTomato* signal" (line 98) doesn't make it clear that the constructs are actually close together on the same chromosome.

More numbers should be provided (i.e. centiMorgans) for how tight the linkage is instead of just describing it as "intimate".

4. The original paper (ref 21) describing Lyn24 as a marker in pollen says that it labels the "Invaginated Pollen Tube Plasma Membrane". Is there a reference that actually calls this structure the "inner vegetative plasma membrane" or have the authors coined a new term? Also, reference 20 does not mention the markers that are described in this part of the manuscript.

5. The authors use some new methods of using dyes (SYBR Green I and Congo Red) stain different features of pollen tubes and ovules. They should introduce what each of these dyes is staining to make it easier for the reader to follow.

6. Fig 3c: please label pollen tubes so that the reader can distinguish the different phenotypes. It is curious that some of the pollen tubes in this image have green nuclei that seem to be far away from the tip. Does SYBER Green affect nuclear movement?

7. Fig 3e: "not determined" might be a better description than "not applicable" in this graph.

8. Fig 4: Can the authors tell if the ovules in the phenotypic classes are fertilized or unfertilized? Is there any evidence of embryo or endosperm development in "A", for example?

9. Fig 6a: Should be "semi in-vitro", not "simi in-vitro"

10. Materials and Methods: Please provide details for the SYBR Green experiment (concentration in the media, microscopy conditions, etc).

Overall, this is a nice manuscript that I enjoyed reviewing.

--Sharon Kessler

Responses to Reviewers

Thank you for reviewing our manuscript entitled “Persistent directional growth capability in *Arabidopsis thaliana* pollen tubes after nuclear elimination from the apex”. By reflecting your constructive comments, our manuscript has substantially changed. We would like to express our sincere gratitude for all the reviewers. The revised version includes new figures: 1 Figure, 4 Supplementary Figures, and 2 Supplementary Movies. The edited texts are highlighted in blue. Details are shown in the followings.

Response to Reviewer #1 comments:

The authors show in this manuscript an interesting study about pollen tube growth and the position of the nuclei. The main contribution of this study is to report a normal pollen tube growth but containing sperm and vegetative basal. It demonstrates that the apical position of the nuclei is dispensable for pollen tube growth. To observe this innovative result, the authors perform interesting experiments by analysing transgenic plants. The experiments have been conducted rigorously with enough replications and support the conclusion. The authors evaluate anomalous callose deposition around sperm cells by making the transgenic line SC-cal containing a nuclear marker. The double mutant *wit1wit2*, previously studied by Zhou and Meier 2014, has also been evaluated in this manuscript, and together with SC-cal, the authors have observed the unusual position of the nuclei, but maintaining normal pollen tube growth. This result is relevant for the study of the mechanisms involved in pollen tube growth and fertilization of flowering plants. I only have a few suggestions and minor points that would help the readability of the paper.

Point 1: Line 101. It would be more accurate to say ‘To reveal sperm cells..’

Response 1: We revised the text according to the comment (lines: 106–109).

Point 2: To further clarify the figure 2, the authors may add SC-cal – and SC-cal + in RHT – and RHT + respectively or the transgenic line carrying the RHT and *pHTRR10::cals3m* can also be named SC-cal, as in the Fig.5 (line 585). The indication of the schematic representation in Fig2a should be more accurate.

Response 2: According to the comment, we indicated the genotype of the pollen or pollen tube as "*SC-cal*⁻, *RHT*⁻" and "*SC-cal*⁺, *RHT*⁺" deduced from the tdTomato signal in Figure 2a, d, and g. In the initial submission, pollen tubes from transgenic plants carrying *SC-cal* and *RHT* were abbreviated as *SC-cal* in Figure 5 to simplify the description of the genotype but were not in the other figures. To provide consistent nomenclature throughout the text, we clarified *RHT* in the genotype of pollen grains, pollen tubes, and plants in the revised manuscript.

Point 3: The callose deposition is clear in the super-resolution image, but have not been included in the schematic representation of *RHT*⁺.

Response 3: We emphasized the callose deposition in Figure 2a.

Point 4: Line 124. Is this transgenic plant the same one that the transgenic line in the fig 2a? The authors should indicate this in the text for clarity, as the authors describe for the *SC-cal RHT* transgenic plant of Fig2a, Fig. 4 and Fig. S3 (line 667–668).

Response 4: Yes. We used the same transgenic line (*SC-cal RHT* hemizygous plants) in Figs 2a, 3b–g and 4. We referred to this in the revised manuscript (lines: 104–105, 130–131, 677–678, and 696–697).

Response to Reviewer #2 comments:

This manuscript by Motomura and colleges describes observations of a series of transgenic *Arabidopsis thaliana* lines to explore the localization of nuclei in growing pollen tubes. Based on their observations, the authors conclude that growth and guidance of pollen tubes can occur independently of a tip-localised vegetative nucleus. This is a novel idea and differs from the general dogma in the area that transcription/translation in the vegetative cell during pollen-tube growth is necessary for continual tip growth and communication with the female tissues. These main conclusions and focus (title, abstract) are drawn from pollen without any nuclei in the pollen-tube tip, but most of the manuscripts focusses on

pollen tubes with altered sperm-cell position with the data on the enucleate pollen being brief and a little preliminary.

The majority of the manuscript describes the impact of ectopic callose expression in sperm cells (SC-Cal). The authors conclude that the SC-Cal sperm cells are no longer tethered to the vegetative cells and are commonly not transported in the pollen-tube tip. The data generally supports these conclusions, and the SC-Cal plants could be a useful tool to study physical interaction between the sperm cells and the vegetative nucleus.

Point 1: I do however have one note on this part of the work. The authors conclude that “ectopic callose deposition on sperm cells caused male sterility” (line 94) and refer to male sterility several times through the manuscript. The data presented does not support this with Fig. 1d indicating up to a 40% transmission efficiency of the transgene (assuming a hemizygous parent) through the male in reciprocal crosses. This shows reduced transmission, not sterility. The manuscript should be edited to reflect this, and the interpretation of the data in the following sections take this into account (e.g. Fig. 4).

Response 1: We changed the word "sterility" to "partial sterility" (lines: 100, 129, 170, and 650).

Point 2: The authors then combine the SC-Cal construct with the previously published *wit1/2* double mutant that impacts vegetative-cell movement into the pollen tube, with the authors confirming the *wit1/2* phenotype. In vitro grown pollen tubes from the SC-Cal *wit1/2* plants were analysed, but the description of the results appears inconsistent with the data presented, potentially as it is unclear what was analysed (all pollen from a *wit1/2* plant hemizygous for SC-cal or only the pollen with the RHT marker linked to SC-cal?).

Response 2: In Figure 5e and f, we showed the analysis of only the tdTomato-positive pollen tubes from the transgenic plants hemizygous for the *RHT* marker linked to *SC-cal*. As mentioned in the response to Reviewer #1, we did not include *RHT* in the genotype descriptions in Figure 5, which we agree was inconsistent throughout the text. We changed those genotype descriptions in the figures and legends to avoid any potential misunderstanding.

Point 3: Line 193 of the text states “In *wit1/2* double mutant plants hemizygous for the *SC-cal* and *RHT* with strong genetic linkage, we observed that vegetative nucleus and sperm cells were isolated in the basal region and the apical region contained no nucleus in 85-94% of the pollen tubes (Fig. 5e,f *SC-cal wit1/2*, *VN&SN basal*)”. But this differs from the data in Fig. 5f where the red bars labelled as *VN&SN basal* look to be ~50 to 75% of the total. Additionally, in Fig. 5f the *SC-cal wit1/2* pollen tubes had an apical vegetative nucleus (*VN&SN apical* and *VN apical*, *SN basal* combined), thus suggesting a substantial amount of the *SC-cal wit1/2* pollen-tube tips do have at least one nucleus. As there is a range of phenotypes in the *SC-cal wit1/2* pollen, it needs to be clearer what was analysed and caution needs to be taken when analysing data from these plants as a population (Fig. 5 h to j).

Response 3: We corrected the frequencies of the *wit1/2 SC-cal RHT* pollen tubes whose vegetative nucleus and sperm cells were isolated in the basal region (Fig. 5f, line: 206-207). As for the phenotype interpretation of the pollen tubes with a range of nuclear positions, we agree with the reviewer’s comment. To draw attention this matter, we mentioned the phenotype variation in the revised manuscript (lines: 253–254).

Point 4: The authors then describe an elegant system to test the growth and guidance of a single pollen tube with a known genotype. This provides some evidence that enucleate pollen tubes can be guided to an ovule, with the authors reporting guidance of pollen tubes with an observed basal vegetative nucleus and sperm cells. However, considering that this is the key conclusion of the paper and there needs to be confidence in this data, but I have a few concerns. Why are the *RHT+* pollen tubes compared only to *RHT-* pollen tubes in Fig. 6e and only to *WT* pollen tubes in Fig. 6f? The authors use this data to conclude there is no impact of pollen-tube growth (6e) and guidance (6f) in *RHT+* plants, but the opposite interpretation could be made if the genotype being compared were reversed, i.e. reduced guidance of *RHT+* pollen tubes (~29%) compared to *RHT-* pollen tubes (~41%). I also wonder about the reliability of the assay, and whether placement of the ovules is such that a certain number of pollen grains will grow to the ovule. It would therefore be good to know how the guidance of *RHT+* tubes compares to tubes with impaired guidance (a guidance mutant), if

the results are statistically significant and how reproducible the assay is.

Response 4: We carried out the single pollen-tube guidance assay using wild-type Col-0 ovules or ovules from the *myb98* homozygous plant that was reported to show a severe defect in ovular guidance of pollen tubes *in vivo*. We found that none of the wild-type pollen tubes penetrated the *myb98* ovules, whereas 20.4% of the wild-type ovules could receive wild-type pollen tubes (Fig. S7, lines: 274–278), confirming the reliability of the single pollen-tube guidance assay. We also performed statistical tests to compare wild-type pollen tubes (Col-0), tdTomato-negative pollen tubes (*wit1/2 SC-cal- RHT-*), and tdTomato-positive pollen tubes (*wit1/2 SC-cal + RHT +*) (new Fig. 7e,f in revised manuscript). There were no significant differences in most of the comparisons shown in Fig. 7e nor all combinations shown in Fig. 7f, and we rephrased the text according to the results (lines: 284–291). Significant differences were detected only between Col-0 and tdTomato-negative pollen tubes (Fig. 7e). However, we did not include this in the revised manuscript, because discussion about the reduced rate of style penetration in the tdTomato-negative *wit1/2* pollen tubes interrupts the main context of the manuscript regarding the characterization of enucleated pollen tubes.

Point 5: The idea that the vegetative nucleus, and notably transcripts from the vegetative nucleus, are not required for normal pollen-tube growth and guidance to ovules differs from previous reports, as discussed briefly in the conclusion. The authors speculate that this is because there is sufficient transcript abundance before the nuclei are sealed off from the pollen-tube tip by a callose plug. It would be great if there was some experimental evidence to support this. While RNA localisation is difficult, such an experiment could involve a fluorescent marker for a pollen-tube expressed gene to confirm the protein is still present (and quantitate the amount) in a pollen-tube tip with a basally-located vegetative nucleus and sperm cells, or RT-qPCR analyses on pollen tubes after emergence for a cut style from the different genotypes. While these experiments may not be necessary, they would add depth to the manuscript.

Response 5: To elucidate the reason anucleate pollen tubes retained their ovule-targeting activity, we quantified mRUBY2-labelled PRK6. PRK6 is a tip-localized receptor-like kinase that perceives AtLURE1 family pollen tube

attractant peptides. Even 8 h after pollination, the fluorescence signal of PRK6-mRUBY2 in enucleated *wit1/2 SC-cal RHT* pollen tubes was evident and the fluorescence intensity was comparable to that in the tdTomato-negative pollen tubes segregated from the same plant (Fig S6 and lines: 261–267).

Minor corrections

Point 6: For Figure 4 the legend states that 48 ovules from nine pistils were classified into seven patterns. Representative images from the nine classes are shown, but it would be best to also show the numbers in each class.

Response 6: We added the number of cases observed to Fig. 4. Additionally, we corrected the frequencies of the phenotypes in Fig. 4d and 4f, because they were swapped in the initial version.

Point 7: Be consistent with nomenclature around *Arabidopsis* or *Arabidopsis thaliana* (currently a mix).

Response 7: We changed the word "Arabidopsis" in the text to "*Arabidopsis thaliana*" or "*A. thaliana*".

Point 8: Please provide AGIs for all genes used.

Response 8: We described the AGIs of all genes in the Methods section (lines: 350–355).

Point 9: Genetic constructs are generally written as promoter:coding region (rather than promoter::coding region).

Response 9: We changed "::" to ":" in the text.

Point 10: When "pollen tube" is being used as a descriptor it should be hyphenated (e.g. pollen-tube growth, pollen-tube function).

Response 10: We revised them in the text.

Point 11: Abstract Line 26: change to “In the pollen tube, the vegetative...”.

Response 11: We changed the text.

Point 12: Line 90: change to “hygromycin-resistance in the F1...”.

Response 12: We revised it.

Point 13: Line 509: Figure 1 legend change “in the sperm cell” to “in sperm cells”.

Response 13: We changed the text.

Point 14: Line 512 Add “plants” to “Aniline blue staining of mature pollen from wild-type (WT, a) or hemizygous pHTR10::cals3m (SC-cal, b) plants.”

Response 14: We added it.

Point 15: Line 513 should be “Transmission of SC-cals ...”.

Response 15: We changed the acronym to uppercase.

Point 16: Relabel the y axes Fig 6e to Pollen tubes emerged from style.

Response 16: We revised it in the new Fig. 7e.

Point 17: Line 221: Remove the s from “Discussions”.

Response 17: We revised it.

Point 18: Please provide a reference for the sentences on lines 222 to 224 and lines 256-257.

Response 18: We added a reference (line: 300). We also revised the order of sentences around lines 256–257 to clarify the text (lines: 313–320).

Point 19: References: The titles of journal articles are in a mix of title case and

sentence case.

Response 19: We corrected the case of the references.

Response to Reviewer #3 comments:

In this exciting manuscript, the authors challenge the long-standing assumption that the positioning of the vegetative nucleus at the tip of the pollen tube is necessary for pollen tube function. They use some clever genetic tools to generate mutant pollen tubes whose sperm cells and vegetative nucleus stay near the pollen grain and show that these pollen tubes can still grow and respond to attractant signals emitted from the ovules. The data are of generally high quality and the results will be of great interest to the plant reproduction community and scientists interested in cell biology, nuclear movement, and intercellular communication.

Point 1: The main takeaway message from the paper is that the vegetative nucleus and sperm cells don't need to be at the tip for the pollen tube to be able to find ovules. The authors try to take the conclusions one step further with the "physiologically anuclear" concept. I think that this description/terminology is based on an assumption that callose plugs are normal and that the tip of the pollen tube has cytoplasm that is isolated from the nuclei that are stuck at the top of the pollen tube. The authors should consider the possibility that apical positioning of the male germ unit is required for normal callose plug formation as the pollen tube grows. If the mutants have abnormal callose plugs, then the pollen tube could be a big continuous cell even when the male germ unit is not at the tip. The authors should investigate these pollen tubes more if they want to claim that the pollen tubes are physiologically anuclear. Some additional experiments to quantify callose plugs (number and spacing) and cytoplasmic streaming in the mutants would provide evidence that these pollen tubes are behaving as anucleate cells.

Response 1: According to the reviewer's comment, we included three figures and a movie. In Supplementary Figure 4, we show time-course observations of *in vitro*-germinated pollen tubes. We found that the *wit1/2 SC-cal RHT* pollen tubes containing immotile sperm nuclei and a vegetative nucleus could produce

callose plugs under normal timing and patterning as well as wild-type Col-0 pollen tubes. (Figure S4, lines: 215–222). Furthermore, a photobleaching experiment of the *wit1/2 SC-cal RHT* pollen tubes expressing cytosolic mTurquoise2 (new Figure 6 and Supplementary Movie 4, lines: 234–248) and the electron microscopy of the cross-sections of *wit1/2 SC-cal RHT* pollen tubes demonstrated the integrity of the first callose plug isolating the immotile sperm cells and vegetative nucleus from the tip-growing apical region (Figure S5, lines: 224–233). We believe these data provide sufficient evidence of the physiologically enucleated condition in the apical region of *wit1/2 SC-cal* pollen tubes.

Point 2: The single pollen tube experiments presented in Figure 6 clearly show that the mutant pollen tubes with the male germ unit stuck near the pollen grain can still be targeted to ovules. It is interesting that the *SC-cal* seems to act as an enhancer of the *wit1/2* phenotype since the main phenotype of *wit1/2* mutants is reversed order of sperm and vegetative nuclei and the *wit1/2* mutants only rarely have a basal VN unless combined with *SC-cal*. Does this suggest that the sperm cells actually “hold” the VN in place and prevent its movement in this triple mutant? It is also interesting that the sperm cells tend to separate from the VN in *SC-cal*, but not in the triple mutant (unless they are actually not connected when they are all basal). The authors should consider adding some more discussion on the influence of the sperm on VN movement when they discuss the sperm having their own motive force.

Response 2: We discussed in more detail the transport of the sperm cells and vegetative nucleus. The apical transport defect of the vegetative nucleus in the *wit1/2* mutant seemed to be enhanced by *SC-cal*, and this could be explained by the tethering of the sperm cells in the basal region or loss of motility in the sperm cells. In the revised manuscript, we added a time-lapse movie of *in vitro*-germinating pollen grains from the *wit1/2 SC-cal RHT* hemizygous plants (Supplementary Movie 3). In this movie, the sperm nuclei and vegetative nucleus were moving around in the grain, which would exclude the tethering hypothesis (lines: 208–212).

We were also aware that *SC-cal*-induced vegetative nucleus–sperm nuclei disconnection was greatly reduced by the *wit1/2* double mutation. This can be partially deduced from the data in Figure 5f, and there are some

additional data to support this interesting effect. However, presenting all the data may disrupt the take-home message regarding the enucleated pollen tube. Thus, we prefer to report the phenomenon in a future paper.

Minor points

Point 3: 1. Line 79: The rationale for expressing *cal3m* in sperm cells might be difficult for non-experts to understand. The concept that sperm cells don't normally have a cell wall and that pollen tubes have a lot of callose in their cell walls and make callose plugs should be included in the introduction or in the leadup to this experiment.

Response 3: We described the callose in the pollen tube, callose plugs, and sperm cells in the introduction (lines: 41–50).

Point 4: 2. The transmission analysis in Fig 1d suggests that *SC-cal* can be transmitted, but only hemizygous mutants are used in the experiments throughout the paper. The authors should comment on why hemizygous mutants were used (i.e. is it possible to get homozygous progeny?)

Response 4: Actually, we obtained homozygous progeny in some *SC-cal* lines. However, we did not have enough time or human resources to prepare homozygous plants in all the lines shown in Figure 5. Hemizygous plants also have an advantage in the analysis of haploid tissues like pollen. Hemizygous plants do not only produce the transgene-present pollen, but do generate the same number of transgene-absent pollen that can be used as a good negative-control (see Fig. 1d,f).

Point 5: 3. The genetic linkage between *SC-cal* and *RHT* should be introduced the first time this genotype is used. The description “cosegregation of callose accumulation and nuclear *tdTomato* signal” (line 98) doesn't make it clear that the constructs are actually close together on the same chromosome. More numbers should be provided (i.e. centiMorgans) for how tight the linkage is instead of just describing it as “intimate”.

Response 5: To select transgenic lines displaying the strong genetic linkage between *SC-cal* and *RHT*, the 1:1 segregation of the tdTomato-positive and tdTomato-negative pollen was confirmed. Then, we analysed whether the presence or absence of the tdTomato signal corresponded to the aniline blue signals on the sperm cells in more than 90% of those pollen. This criterion was shown in the Methods section (lines: 411–415).

Point 6: 4. The original paper (ref 21) describing Lyn24 as a marker in pollen says that it labels the “Invaginated Pollen Tube Plasma Membrane”. Is there a reference that actually calls this structure the “inner vegetative plasma membrane” or have the authors coined a new term? Also, reference 20 does not mention the markers that are described in this part of the manuscript.

Response 6: The nomenclature of the sperm cell-enclosing membrane is not fixed even today. For example, a recently opened preprint used “Pollen endo-plasma membrane” (<https://doi.org/10.1101/2020.10.05.326157>), but this nomenclature was not used elsewhere. As mentioned by the reviewer, the paper reported the Lyn24-GFP described the membrane as “invaginated pollen tube plasma membrane (IPTPM)” (<https://doi.org/10.1093/mp/sst098>). However, this nomenclature can be used only after the germination of pollen tube. After an extensive literature search, we finally found some old papers that used the term “inner vegetative plasma membrane” (<https://doi.org/10.1007/BF01332646>, [https://doi.org/10.1016/0168-9452\(94\)90184-8](https://doi.org/10.1016/0168-9452(94)90184-8)). In our manuscript, we wanted to discuss the function and behavior of the sperm cell-enclosing membrane before and after pollen-tube germination, and “inner vegetative plasma membrane” was the best term for our purpose. As for the reference number, we corrected the mistake.

Point 7: 5. The authors use some new methods of using dyes (SYBR Green I and Congo Red) stain different features of pollen tubes and ovules. They should introduce what each of these dyes is staining to make it easier for the reader to follow.

Response 7: We added the information to the text (lines: 137–138, 156, 681, and 694).

Point 8: 6. Fig 3c: please label pollen tubes so that the reader can distinguish the different phenotypes. It is curious that some of the pollen tubes in this image have green nuclei that seem to be far away from the tip. Does SYBER Green affect nuclear movement?

Response 8: We indicated the *SC-cal + RHT+* pollen tubes with asterisks (Fig. 3d). In SYBR Green I-labelled pollen tubes, vegetative nuclei were ~70 μm from the tip (Fig. 3d, Supplementary Fig. 2b). This was relatively longer than *tdTomato*-expressing pollen tubes in the *RHT* hemizygous lines in Figure 5i (mean of the Tip-VN ranged from 40.8 μm to 51.4 μm in the three independent lines). However, we could not conclude that phenotypic difference was caused by SYBR Green I because the experimental conditions in Fig. 3 were considerably different from those in Fig. 5 (e.g. semi-*in vitro* germination or *in vitro* germination, recipes of the growth media, etc.).

Point 9: 7. Fig 3e: “not determined” might be a better description than “not applicable” in this graph.

Response 9: We revised the figure.

Point 10: 8. Fig 4: Can the authors tell if the ovules in the phenotypic classes are fertilized or unfertilized? Is there any evidence of embryo or endosperm development in “A”, for example?

Response 10: In Figure 4, the fertilization phenotype was unclear. The *pRPS5A:H2B-tdTomato (RHT)* marker can label the embryo- and endosperm-nuclei in the developing seeds when it is inherited from the male parent (Maruyama et al., 2013, <http://doi.org/10.1016/j.devcel.2013.03.013>). However, the shortage of *RHT* is a late signal increase after double fertilization. Although *pRPS5A:H2B-GFP (RHG)* showed rapid increase in the GFP signal within 2 hours after fertilization, the signal elevation in *RHT* was not evident even after 6 hours after fertilization. This may reflect the different folding time of those proteins. In Figure 4, we observed the ovules at 16 hours after pollination. At this stage, the nuclear signal in the embryo and endosperm was still weak, and the elevated background signal by the Congo Red made it more difficult to distinguish the fertilization phenotype. We thus only showed the quantitative

data that counted the punctate tdTomato signal, most of which likely corresponded to the vegetative nucleus.

Point 11: 9. Fig 6a: Should be “semi in-vitro”, not “simi in-vitro”

Response 11: We corrected the term.

Point 12: 10. Materials and Methods: Please provide details for the SYBR Green experiment (concentration in the media, microscopy conditions, etc).

Response 12: We described SYBR Green I in the “*In vitro* and semi-*in vitro* pollen-tube growth” section in the Methods.

Overall, this is a nice manuscript that I enjoyed reviewing.
--Sharon Kessler

Response: It is a great pleasure to hear that. Thank you.

REVIEWERS' COMMENTS

Reviewer #2 (Remarks to the Author):

The original version of this manuscript raised an interesting question about the role of the vegetative nucleus in growing pollen tubes of Arabidopsis, suggesting that the vegetative nucleus does not need to be located in the growing apex. My main concern with the original manuscript was that the data supporting this was preliminary, with many assumptions being made. In the revised manuscript, the authors have drawn on further genetic resources to add new images and data that satisfy my concerns, and add depth to the story. Overall the manuscript is logically organised and well presented with some beautiful figures and movies, and causes a rethink of the dogma in the field. Reading the manuscript still led to me ask a lot of questions about the mechanism of how pollen-tube growth then works (timing of transcription and transport of mRNA). However, I feel answering these is beyond the scope of what would be required for this manuscript and instead should be seen as a strength of the manuscript; it will leave readers thinking and open up new experimental work in the area.

While overall, the manuscript presents an interesting story and the data supports the conclusions made, I still have some relatively minor questions and some suggestions to increase the accessibility and readability for those outside the field and improve overall flow.

Line 31 "Here, we succeeded in generating pollen tubes in..." to "We generated pollen tubes in..."

Line 32 In the sentence "...whose vegetative nucleus and sperm cells were isolated in the plugged basal region because..." the term "plugged" may be confusing to readers outside the field. Could be changed to "...whose vegetative nucleus and sperm cells were isolated and sealed by callose plugs in the basal region because...".

Lines 34/35 : "CALLOSE SYNTHASE 3 mutant" could be changed to a "dominant mutant of the CALLOSE SYNTHASE 3 protein".

Lines 53-55. The term "male germ unit" only needs to be introduced once.

Line 79. Removed the "s" from "positions" in "vegetative nucleus positions".

Line 82. Potentially change "cell wall" to "cell periphery" as it has previously been stated that sperm cells do not have a wall (line 50).

Line 87. It would be good to know what the gain-of-function mutation causes (e.g. constitutive activity?)

Line 94, Fig1 c. The text notes that four lines have ~50% hygromycin resistant seedlings, while the figure also shows a fifth line with a higher percentage. This is probably due to two inserts of the transgene, but it would be good to note this in the text or figure legend.

The changes to the nomenclature and the added descriptions concerning the SC-cal RHT hemizygous lines has clarified the use of this line. However, it is still not clear the first time this is shown in Fig. 2a (and it is easy to miss the "co-segregation" mentioned on line 103". It would be beneficial if the co-segregation or genetic linkage was shown on the schematic in Fig 2a and the term used consistently.

Line 119. It would be good to explain what the function of the N-terminal region of the Lyn protein is.

Line 120. Please check the reference. In the paper referenced (22) the KRP6 promoter is used for vegetative-cell expression and the ACA3 promoter is not mentioned. It is important to have strong verification that this is a vegetative cell-specific promoter, as in the images shown it is difficult to determine if the labelled membranes are of sperm cell or vegetative cell origin, but the authors are using the construct to identify the inner vegetative cell plasma membrane.

Line 122. "observed in 76.9% of the pollen of tdTomato-negative devoid of SC-cal" is unclear. Could be "observed in 76.9% of the SC-cal- RHT- pollen".

I am a little confused by the data in Supplementary Fig 2. The data indicates that there is no significant difference in the distance between the tip-VN, and the tip-SN in SC-cal- RHT- and SC-cal+ and RHT+ pollen tubes, but there is a difference between the VN-SN. As the Tip-SN is the combination of the Tip-VN, the VN and the VN-SN (as labelled on part a) it is hard to see how the total (Tip-SN) stays the same when one of the components (VN-SN changes). Additionally, I assume these measurements were only taken in tubes where the SNs had entered the pollen tube. This should be made clear in the legend.

Line 146-148. Please include the genotype being studied in this sentence.

Lines 158-161. It would be good to explain the logic here for non-expert readers, in particular why a lack of td-Tomato signals indicates reception. Further, it would also be good to label the pollen tubes in Fig 4 for clarity.

Line 178. Add "positioned" between "nucleus" and "ahead".

Line 185. Change "inhibited transport to the apical region of the sperm cells" to "inhibited transport of the sperm cells to the apical region of the pollen tube".

Line 197. Change "orientation of the unit was inverted to form the sperm nuclei first order in 93-96%" to "orientation of the unit was inverted with the sperm nuclei first in 93-96%".

Line 213. This seems to be stating the conclusion before data is shown. It could be stated that this was a hypothesis to be tested, or otherwise edited.

Line 218. Change "immotile nuclei generated the first callose plug within 3 h after germination as well as the Col-0 wild-type pollen tubes" to "immotile nuclei generated the first callose plug within 3 h after germination, which was similar in Col-0 wild-type pollen tubes"

Lines 221 to 222. I am a little confused by the conclusion "suggesting that the positions of the sperm cells and vegetative nucleus are independent of callose-plug formation." Should it be the other way around? i.e. "suggesting that the deposition of the callose plugs is independent of the position of the sperm cells and vegetative nucleus"?

Line 223. "suggesting the integrity of the first callose plug" seems incomplete. "suggesting the integrity of the first callose plug is not altered" or "suggesting the first callose plug is completed".

Lines 246-248. "supporting the integrity of the first callose plug that prevents acropetal and basipetal material transport in the pollen tube containing immotile sperm cells and the vegetative nucleus" is a little difficult to follow. "This indicates that the presence of the callose plug is preventing the movement of the mTurquoise2 protein from the basal regions to the growing regions of the pollen tube, and that potentially the movement of other molecules is also prevented".

Line 249. It would be good to define a little more what is meant by "physiologically enucleated condition".

I think the data shown in Supplemental Fig 6 are important in showing that the pollen-tube tip still contains high levels of proteins in supporting the hypothesis that the pollen-tube growing regions has inherited sufficient transcripts from the pollen grain/ young tip to sustain growth. I would consider putting them in the body of the text. It would also be useful to state the expression pattern of these promoters (LAT52 and PRK6), e.g. the mRNA for both of these is expected to be present in the mature pollen grain.

Line 276. Please clarify if the statement "pollen tubes entered 20.4% of ovules" is the right way around? Or did 20% of pollen tubes enter an ovule? Looking at the assay with three ovules and a single pollen tube, then only 33% of ovules would be entered. Also in line 288 "occurred in 28.6% of" could be "occurred with 29% of".

Line 306. Change "could induce apical transport defect in sperm cells" to "could induce defects in the apical transport of sperm cells".

Line 330. Add the word likely i.e. "formation would likely interrupt".

Line 359. Please add if the mutants were also in to Col-0 background.

Line 364 and elsewhere. Please note where the upstream region is taken from (e.g. the translation initiation site/ATG).

Line 433. Replace "coverslipped" with "covered by a coverslip" or something similar.

Fig. 1 legend. Change to "Partial male sterility induced".

Fig. 2 legend. Please provide more detail for panel a, particularly around the two inserts being linked along with a description of the schematic shown below. Additionally add "pollen from" on line 668 (i.e. NeonGreen, in the pollen from a SC-cal..).

Fig. 3 and Fig. 5 legends. Change "wild-type pollen" to "pollen tubes from a plant transformed with only..." or "single transgenic". For Fig.3 it would be nice to describe the schematic in panel c in more detail.

Fig. 6 legend, line 726. Change "immobile" to "basal".

Reviewer #3 (Remarks to the Author):

The authors did a good job of addressing my concerns, especially with regards to the callose plug phenotypes. They also gave satisfactory responses to the other reviewers' comments.

Responses to Reviewers

Thank you for reviewing our manuscript entitled “Persistent directional growth capability in *Arabidopsis thaliana* pollen tubes after nuclear elimination from the apex” and for your constructive comments and suggestions. Please see our responses below.

Response to Reviewer #2 comments:

Line 31 “Here, we succeeded in generating pollen tubes in...” to “We generated pollen tubes in...”

Response: Thank you for your suggestion. We have revised the sentence accordingly.

Line 32 In the sentence “...whose vegetative nucleus and sperm cells were isolated in the plugged basal region because...” the term “plugged” may be confusing to readers outside the field. Could be changed to “...whose vegetative nucleus and sperm cells were isolated and sealed by callose plugs in the basal region because...”.

Response: We revised the word “plugged” to “sealed” in the revised manuscript (Lines: 33).

Lines 34/35 : “CALLOSE SYNTHASE 3 mutant” could be changed to a “dominant mutant of the CALLOSE SYNTHASE 3 protein”.

Response: We have rephrased the text accordingly (Lines: 36).

Lines 53-55. The term “male germ unit” only needs to be introduced once.

Response: We have removed redundant expressions from the revised manuscript (Lines: 56).

Line 79. Removed the “s” from “positions” in “vegetative nucleus positions”.

Response: We have changed it accordingly (Lines: 81).

Line 82. Potentially change “cell wall” to “cell periphery” as it has previously been stated that sperm cells do not have a wall (line 50).

Response: We used “periphery” in the revised manuscript (Lines: 85).

Line 87. It would be good to know what the gain-of-function mutation causes (e.g. constitutive activity?)

Response: The cause of callose over-accumulation by the *cals3m* remains to be determined. However, a previous study has suggested the possibility of constitutively active mutation in which the point mutation of *cals3m* may induce a loss of negative regulation of the callose synthase activity. We therefore introduced the *cals3m* mutant as “putative constitutively-active mutant of CALLOSE SYNTHASE 3 (*cals3m*)” in the revised manuscript (Lines: 89–90).

Line 94, Fig1 c. The text notes that four lines have ~50% hygromycin resistant seedlings, while the figure also shows a fifth lines with a higher percentage. This is probably due to two inserts of the transgene, but it would be good to note this in the text or figure legend.

Response: We mentioned the relatively high hygromycin resistance in line 2 of the figure legend (Lines: 679–681).

The changes to the nomenclature and the added descriptions concerning the SC-cal RHT hemizygous lines has clarified the use of this line. However, it is still not clear the first time this is shown in Fig. 2a (and it is easy to miss the “co-segregation” mentioned on line 103”. It would be beneficial if the co-segregation or genetic linkage was shown on the schematic in Fig 2a and the term used consistently.

Response: We revised the diagram to show the co-segregation of *SC-cal* and *RHT* in Figure 2a and used the term “genetic linkage/genetically-linked” consistently.

Line 119. It would be good to explain what the function of the N-terminal region of the Lyn protein is.

Response: *Lyn24* has the targeting signals of both myristoylation and palmitoylation, and is essential for the inner vegetative plasma membrane localization. We have added this to the revised manuscript (Lines: 124–126).

Line 120. Please check the reference. In the paper referenced (22) the KRP6 promoter is used for vegetative-cell expression and the ACA3 promoter is not mentioned. It is important to have strong verification that this is a vegetative

cell-specific promoter, as in the images shown it is difficult to determine if the labelled membranes are of sperm cell or vegetative cell origin, but the authors are using the construct to identify the inner vegetative cell plasma membrane.

Response: The vegetative cell-specific gene expression from the *ACA3* promoter is shown in supplementary figure S1 of Reference 22. In this paper, the *ACA3pro::H2B-GFP* reporter exclusively labeled the vegetative nucleus only after the tricellular stage. Some of the major vegetative cell-specific promoters (e.g., *LAT52* promoter) showed weak gene expression in sperm cells. We therefore used the *ACA3* promoter for the expression of Lyn24-mNeonGreen to ensure that the eye glass-shaped mNeonGreen signal is derived from the inner vegetative plasma membrane rather than the sperm cells.

Line 122. “observed in 76.9% of the pollen of tdTomato-negative devoid of SC-cal” is unclear. Could be “observed in 76.9% of the SC-cal⁻ RHT⁻ pollen”.

Response: We have revised it as suggested (Lines: 127).

I am a little confused by the data in Supplementary Fig 2. The data indicates that there is no significant difference in the distance between the tip-VN, and the tip-SN in SC-cal⁻ RHT⁻ and SC-cal⁺ and RHT⁺ pollen tubes, but there is a difference between the VN-SN. As the Tip-SN is the combination of the Tip-VN, the VN and the VN-SN (as labelled on part a) it is hard to see how the total (Tip-SN) stays the same when one of the components (VN-SN changes). Additionally, I assume these measurements were only taken in tubes where the SNs had entered the pollen tube. This should be made clear in the legend.

Response: Thank you for your suggestion and we agree with your assessment. The Tip-SN in the *SC-cal⁺ RHT⁺* pollen tubes appeared to be longer than those in the *SC-cal⁻ RHT⁻* pollen tubes, however, no significant differences were observed for a P value of < 0.01, which is because the VN-only pollen tubes were excluded from the analysis. To avoid confusion, we have referred to sample population of the *SC-cal⁺ RHT⁺* pollen tubes analyzed in the legend of Supplementary Figure 2.

Line 146-148. Please include the genotype being studied in this sentence.

Response: We have added the genotype as suggested (Lines: 152).

Lines 158-161. It would be good to explain the logic here for non-expert readers, in particular why a lack of td-Tomato signals indicates reception. Further, it would also be good to label the pollen tubes in Fig 4 for clarity.

Response: For better clarity, we have revised the sentence and indicated our expected result (i.e., our hypothesis) prior to our observation (Lines: 165–169). We also labeled the pollen tube in Figure 4.

Line 178. Add “positioned” between “nucleus” and “ahead”.

Response: Thank you for your suggestion. We have revised the text accordingly. (Lines: 187).

Line 185. Change “inhibited transport to the apical region of the sperm cells” to “inhibited transport of the sperm cells to the apical region of the pollen tube”.

Response: Thank you for your suggestion. We have revised the text accordingly (Lines: 195).

Line 197. Change “orientation of the unit was inverted to form the sperm nuclei first order in 93–96%” to “orientation of the unit was inverted with the sperm nuclei first in 93–96%”.

Response: Thank you for your suggestion. We have revised the text accordingly (Lines: 207).

Line 213. This seems to be stating the conclusion before data is shown. It could be stated that this was a hypothesis to be tested, or otherwise edited.

Response: We referred to the first callose plug in the *wit1/2 SC-cal RHT* pollen tubes showed in Figure 5e rather than the conclusion in Supplementary Fig. 4. To clarify this, we stated the figure panel and rephrased the sentence (Lines: 224–226).

Line 218. Change “immotile nuclei generated the first callose plug within 3 h after germination as well as the Col-0 wild-type pollen tubes” to “immotile nuclei generated the first callose plug within 3 h after germination, which was similar in Col-0 wild-type pollen tubes”

Response: Thank you for your suggestion. We have revised the text accordingly (Lines: 230).

Lines 221 to 222. I am a little confused by the conclusion “suggesting that the positions of the sperm cells and vegetative nucleus are independent of callose-plug formation.” Should it be the other way around? i.e. “suggesting that the deposition of the callose plugs is independent of the position of the sperm cells and vegetative nucleus”?

Response: Thank you for your comment. We have corrected the statement in the revised manuscript (Lines: 233–234).

Line 223. “suggesting the integrity of the first callose plug” seems incomplete. “suggesting the integrity of the first callose plug is not altered” or “suggesting the first callose plug is completed”.

Response: Thank you for your suggestion. We have revised the text accordingly (Lines: 246).

Lines 246-248. “supporting the integrity of the first callose plug that prevents acropetal and basipetal material transport in the pollen tube containing immotile sperm cells and the vegetative nucleus” is a little difficult to follow. “This indicates that the presence of the callose plug is preventing the movement of the mTurquoise2 protein from the basal regions to the growing regions of the pollen tube, and that potentially the movement of other molecules is also prevented”.

Response: Thank you for your suggestion. We have revised the text accordingly (Lines: 259–262).

Line 249. It would be good to define a little more what is meant by “physiologically enucleated condition”.

Response: Thank you for the suggestion. We have included additional information regarding the physiologically enucleated condition in the revised manuscript (Lines: 264–265).

I think the data shown in Supplemental Fig 6 are important in showing that the pollen-tube tip still contains high levels of proteins in supporting the hypothesis that the pollen-tube growing regions has inherited sufficient transcripts from the pollen grain/ young tip to sustain growth. I would consider putting them in the body of the text. It would also be useful to state the expression pattern of these

promoters (LAT52 and PRK6), e.g. the mRNA for both of these is expected to be present in the mature pollen grain.

Response: Thank you for your suggestion. Supplementary Figure 6 has been added to the revised manuscript as new Figure 7. We also mentioned the expression pattern of *LAT52* and *PRK6* promoters (Lines: 275–276).

Line 276. Please clarify if the statement “pollen tubes entered 20.4% of ovules” is the right way around? Or did 20% of pollen tubes enter an ovule? Looking at the assay with three ovules and a single pollen tube, then only 33% of ovules would be entered. Also in line 288 “occurred in 28.6% of” could be “occurred with 29% of”.

Response: Thank you for your comment. We have corrected the statement in the revised manuscript (Lines: 292–293, 304).

Line 306. Change “could induce apical transport defect in sperm cells” to “could induce defects in the apical transport of sperm cells”.

Response: Thank you for your suggestion. We have revised the text accordingly (Lines: 322–323).

Line 330. Add the word likely i.e. “formation would likely interrupt”.

Response: Thank you for your suggestion. We have revised the text accordingly (Lines: 346).

Line 359. Please add if the mutants were also in to Col-0 background.

Response: We have mentioned the background of the mutants in the revised manuscript (Lines: 374–375).

Line 364 and elsewhere. Please note where the upstream region is taken from (e.g. the translation initiation site/ATG).

Response: We have added the information to the revised manuscript (Lines: 380–381, 394–395).

Line 433. Replace “coverslipped” with “covered by a coverslip” or something similar.

Response: Thank you for your suggestion. We have revised the text accordingly (Lines: 453).

Fig. 1 legend. Change to “Partial male sterility induced”.

Response: Thank you for your suggestion. We have revised the text accordingly (Lines: 673).

Fig. 2 legend. Please provide more detail for panel a, particularly around the two inserts being linked along with a description of the schematic shown below. Additionally add “pollen from” on line 668 (i.e. NeonGreen, in the pollen from a SC-cal..).

Response: We have described the co-segregation of the two transgenes in Figure 2a and added "pollen from" to the text (Lines: 694).

Fig. 3 and Fig. 5 legends. Change “wild-type pollen” to “pollen tubes from a plant transformed with only...” or “single transgenic”. For Fig.3 it would be nice to describe the schematic in panel c in more detail.

Response: Thank you for your suggestion. We have revised the text accordingly (Lines: 700–701, 731). We also revised Figure 3c.

Fig. 6 legend, line 758. Change “immobile” to “basal”.

Response: Thank you for your suggestion. We have revised the text accordingly (Lines: 758).